



# On the Seasonal Western Boundary Current System of the Weddell Gyre

Tania Pereira-Vázquez [a], Borja Aguiar-González [a*], Ángeles Marrero-Díaz [a], Marta Veny [a], Ángel Rodríguez-Santana [a]

[a] Oceanografía Física y Geofísica Aplicada (OFYGA), ECOAQUA, Universidad de Las Palmas de Gran Canaria, Canary Islands, 35017, Spain

*Correspondence to*: Borja Aguiar-González (borja.aguiar@ulpgc.es)

**Abstract.** We investigate the seasonal Western Boundary Current System (WBCS) of the Weddell Sea Gyre using two open-
access global ocean circulation reanalysis products (NEMO and HYCOM at different resolutions), direct velocity measurements and altimetry data across an extended version of the historical ADELIE transect, hereafter E-ADELIE. The NEMO-based products are GLORYS2V4 and GLORYS12V1, provided daily with 0.25º and 0.08° of horizontal resolution, respectively. The HYCOM-based product is GLBv0.08, provided daily with 0.08º of horizontal resolution between 40°S and 40°N, and 0.04º beyond these latitudes. The ADELIE extension is made to include a novel, persistent current, previously
unreported, which we name as the Inner Weddell Current (IWC). With this approach, we aim to assess whether these open-access products capture properly the dynamics, natural mode of variability and spatio-temporal scales of the WBCS so that we can set the groundwork for future interannual variability studies. E-ADELIE is located at a key location of the WBCS, before it splits into different branches that redistribute Weddell Sea waters either leaving the basin towards Bransfield Strait, towards the South Atlantic Ocean, or recirculating within the gyre. The analyses include the characterization of the
horizontal and vertical structure of the WBCS and its volume transport. Results show that both reanalysis products agree on key dynamics features only at high model resolutions; NEMO at the lower-resolution version lacks the typical multi-jet structure of the WBCS. The altimetry data is also in agreement in showing this year-round multi-jet structure but in winter, when data gaps prevent us from a comprehensive view. The cross-transect volume transport variability from the reanalysis products is consistent with the seasonality of the basin scale wind forcing in all model cases, with minimum values (25 ± 5
Sv) from September to December and maximum values (33 ± 5 Sv) from March to July. Also, the time-averaged transport aligns at approximately 30 ± 5 Sv in all products. Major discrepancies exist towards the interior of the gyre, where the IWC differs strongly among reanalysis products. The IWC is found about 50 km east of the Weddell Front extending up to 600 Km offshore, standing out as a bottom-reaching, broad current transporting ~14±3 Sv in NEMO and ~39±5 Sv in HYCOM. According to available Shipboard Acoustic Doppler Current Profiler (SADCP) measurements from two different surveys,
LG0003a and NBP0106, we find the IWC surface volume transport (0-300 m depth) is about 6.67 Sv and 7.64 Sv, respectively between 400 and 720 km offshore the Antarctic Peninsula. When computing the same estimates from the reanalysis products, we obtain lower volume transports values ranging from 2-4 Sv. Results from this study suggest that the high-resolution version of NEMO (GLORYS12V1) approaches the real ocean in the western Weddell Sea the closest when



compared to observations and literature. These results open the avenue for future research investigating the variability of the

WBCS of the Weddell Gyre at interannual scales at a key location for water mass exchange between ocean basins.

**Keywords**: Cyclonic Weddell Gyre, Volume Transport, Western Boundary Current System, Southern Ocean, Global Ocean Circulation Reanalysis Products.

## 1 Introduction

The major feature of the circulation in the Weddell Sea is a cyclonic wind-driven gyre, subject to thermohaline forcing and topographic steering (Absy et al., 2008; Vernet et al., 2019). This gyre is located roughly between 65-78°S and 60°W-20°E (Fig.1). Along its southern boundary, the Weddell Sea is bounded by the Antarctic continent, while along its northern boundary, it is open to the Southern Ocean.

A particularly complex pattern of currents takes place along the Western Boundary Current System (WBCS) of the Weddell Sea. Over this domain, a multi-jet structure is developed and water masses flowing within the western branch of the gyre either recirculate within the gyre or leave the basin towards the Bransfield Strait or the Scotia Sea, then flowing northward into the South Atlantic Ocean (Hellmer et al., 2005; Naveira Garabato et al., 2002). These exported water masses change the Earth's climate by altering the total heat and carbon content in the global ocean (Lumpkin & Speer, 2007; Styles et al., 2023;

Talley et al., 2003; Vernet et al., 2019). Thus, if climate models and global ocean circulation reanalysis products do not properly resolve the Weddell Gyre circulation and converge towards aligned results, long-term projections could become divergent and lead to inconclusive future case scenarios. In this work, we assess the baseline variability of the Weddell Gyre circulation along its western boundary, focusing on seasonal variations, to establish a foundation for future research exploring longer-term variability. Hence, we define austral seasons as follows: summer (January - February - March),

autumn (April - May - June), winter (July - August - September), spring (October - November - December), following the criteria of Zhang et al. (2011) and Dotto et al. (2021).The WBCS of the Weddell Sea presents three major bottom-intensified currents with a remarkable barotropic structure corresponding to the Antarctic Coastal Current (CC), the Antarctic Slope Front (ASF), and the Weddell Front (WF) as shown in Fig. 1 (Kerr et al., 2012; Stewart & Thompson, 2016; Thompson & Heywood, 2008; Thompson el al., 2018).


The CC drives the exit of Weddell waters towards Bransfield Strait, allowing the leakage of near-freezing subsurface waters. In this manner, the CC forms a cold-water pathway which feeds the west Antarctic Peninsula and maintains regionally low rates of glacier retreat (Cook et al., 2016). Farther offshore, the jet associated with the ASF is found. This jet is formed along the continental slope of Antarctica as a strong, narrow and persistent northward-flowing current (Thompson & Heywood,

2008; Vernet et al., 2019). Across this front there is a rapid change in temperature and salinity due to the interaction between



the colder and fresher waters of the Antarctic continental shelf, formed as a result of sea ice formation and melting processes, and the relatively warm and saline waters sourced by the Antarctic Circumpolar Current (ACC) (Thompson & Heywood, 2008; Vernet et al., 2019). The former water mass is known as Antarctic Surface Water (AASW), while the latter water mass is a modified Circumpolar Deep Water known as Warm Deep Water (WDW). Down the continental slope, dense Antarctic

Bottom Water (AABW) is formed following brine rejection in the upper ocean surface. Thus, the ASF plays a crucial role in the exchange of heat, salt, and nutrients between the deep ocean and the Antarctic continental shelf waters with important implications for the distribution of marine ecosystems and sea-ice dynamics, where the WDW also conditions the melting of ice shelves. Lastly, towards the ocean interior, another jet is found associated with the WF, which holds a major frontal system linked to high mesoscale variability in the form of eddies and meanders (Heywood et al., 2004; Thompson &

Heywood, 2008). An overview of the water masses which compose the multi-jet structure of the WBCS in the reanalysis products as compared to observations is provided in the Appendix.

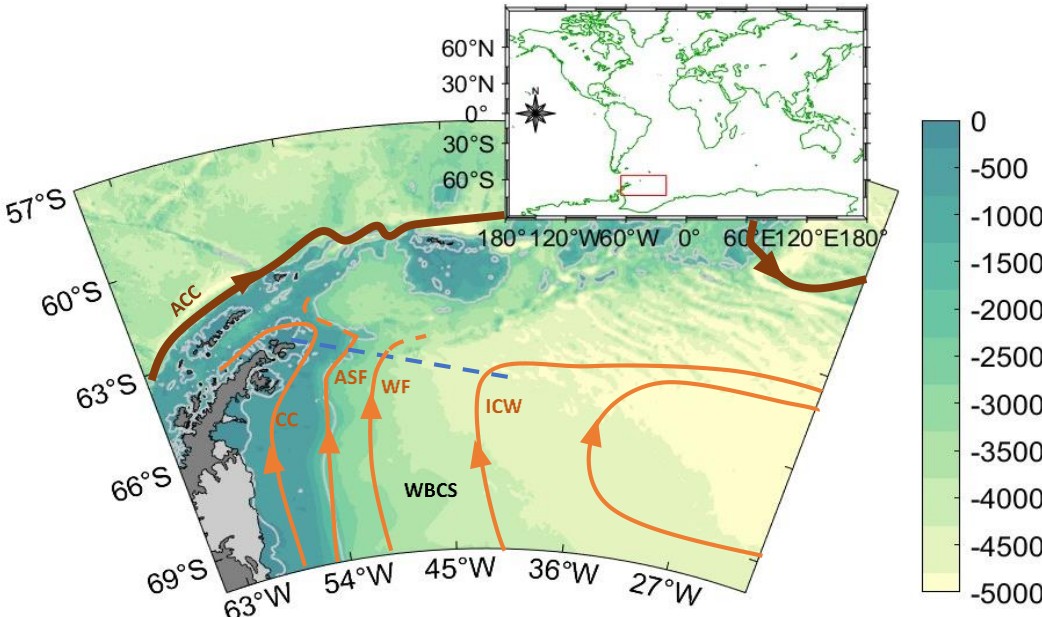

Figure 1. Bathymetric map of the north-western sector of the Weddell Gyre depicting the study area with an indication to the approximate location of major oceanographic features and currents with a surface signal. The E-ADELIE transect localization is indicated in blue (see section 2.2.1). Acronyms stand for: Southern Boundary of Antarctic Circumpolar Current (ACC), Western Boundary Current System

(WBCS), Antarctic Coastal Current (CC), Antarctic Slope Front (ASF), Weddell Front (WF) and Inner Weddell Current (IWC). The latter current is described for the first time in this study.

When considering a basin situated at high latitudes, it is not unexpected to anticipate that the horizontal resolution of the model will most likely play a crucial role in the realistic simulation of the ocean dynamics of the Weddell Sea. Recently, an

idealized model of the Weddell Sea gyre was investigated to test how inter-model variability can originate from differences



in the horizontal resolution of the ocean model (Styles et al., 2023). Through this work, the authors aimed to investigate how mesoscale eddies can influence the Weddell Gyre transport and its interaction with the Antarctic Circumpolar Current (ACC). To do this, they run an idealized model varying the horizontal resolution over a wide range of horizontal grid spacings following eddy-parameterized scales (80 and 40 km), eddy-permitting scales (10 and 20 km), and eddy-rich scales

(3 km), thus accounting for the small Rossby deformation radius at these high latitudes. The output differences among runs were far from neglectable, and the authors concluded that the Weddell Gyre is extremely sensitive to horizontal resolution, displaying the strongest transport at eddy-permitting resolutions (45 Sv), where only the largest ocean eddies are resolved. These resolutions produced unique density structures and substantial thermal wind transport, with the thermal wind component being largest at eddy-permitting resolutions due to the less efficient flattening of isopycnals by the partially-

resolved eddy field. The smallest gyre transport was found at eddy-parameterized resolutions (12 Sv).  Importantly, the authors warned against adopting resolutions that the Weddell Gyre is most sensitive to in state-of-the-art climate models, as it could lead to more divergent simulations. Their recommendation to ocean modelers is to approach the eddy-permitting resolution with care when simulating the Southern Ocean and to consider employing parameterizations that are compatible with partially resolved mesoscale eddies.


It is worthwhile noting that, in the research conducted by Styles et al. (2023), the authors adopted a time-averaged forcing to produce the Weddell Gyre and ACC subject of their study. Styles et al. (2023) acknowledged that this idealized approach poses limitations, as in the real ocean the Weddell Gyre and ACC experience pronounced seasonal fluctuations. These fluctuations significantly alter the gyre's transport (Neme et al., 2021) and the density structure on its western boundary

(Hattermann, 2018). However, estimates of the Weddell Gyre transport based on observations are actually constrained across seasons, particularly during winter, and exhibit considerable variability (readers will find a detailed review about former studies in Section 3, where we discuss our findings with existing literature).

This scenario fuels the motivation behind this work. We do not use an idealized model, neither a climate model but two

global ocean circulation reanalysis products with different horizontal resolutions and a set of in situ and remotely-sensed observations. The target is to build further insights about the seasonal variability of the Weddell Gyre circulation along its western boundary upon the results from previous observational and modelling works. Thus we aim an improved understanding not only of the most characteristic features of the year-round ocean dynamics of the gyre but also of the most likely sensitive factors to account for a realistic representation of the gyre circulation in open-access reanalysis products so

that the scientific community can use them as a reliable tool to explore ongoing interannual and longer-term variability.

In this context, the historical transect known as ADELIE (Antarctic Drifter Experiment: Links to Isobaths and Ecosystems), depicted in Figs. 1 & 2, constitutes the most convenient reference frame to assess the ocean dynamics in the northwest of the Weddell Gyre (Thompson & Heywood, 2008). Estimates of volume transport across this (or analogous) transects have been



reported in the literature to range between 24-29 Sv, depending mainly on the season and the performed sampling strategy
(Gordon, 2020; Muench & Gordon, 1995; Thompson & Heywood, 2008). However, in situ observations reported in the
literature are mostly representative of the free sea-ice seasons since the extreme meteorological conditions prevailing during
the rest of the year prevent us from counting with year-round observations. Also, remotely-sensed observations about the
surface ocean dynamics are strongly limited during the sea-ice seasons because the presence of the sea-ice cover poses a
major challenge for satellite sampling. Hence, the use of reanalysis products becomes crucial so we can explore the
governing year-round ocean dynamics in this region without time gaps. To this aim, we analyse the horizontal and vertical
structure of the WBCS and its volume transport across an extended version of the historical ADELIE transect, which spans
oceanward from the northernmost tip of the Antarctic Peninsula, and across the WBCS towards the interior of the gyre. We
extend the original ADELIE transect, and refer to it hereafter as the Extended ADELIE transect (E-ADELIE, for short, in
Fig. 2) to account for the full extent of the circulation across the western Weddell Gyre.

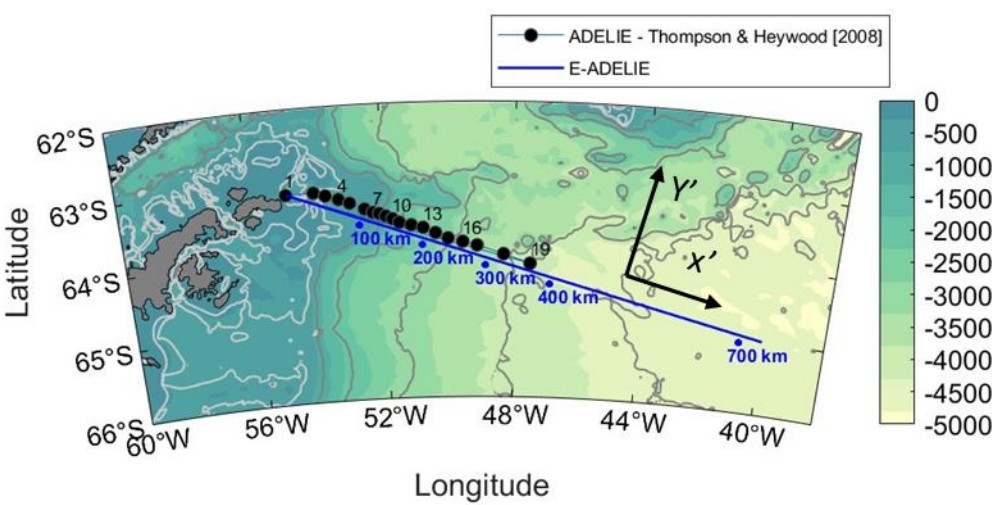

Figure 2. Bathymetric map of the north-western sector of the Weddell Gyre depicting the study area with indication of the ADELIE
transect (Cruise 158 of RRS James Clark Ross (JCR) in February 2007) studied in Thompson & Heywood (2008), and the E-ADELIE
transect we study. The ADELIE stations are indicated with black dots; the station numbers are also shown. Distance of E-ADELIE is
indicated along the transect. The new frame of reference for the velocity field is also indicated (the Cartesian coordinates were rotated
15.79 degrees clockwise ($\alpha$) from the true east following Eq. 1). Fine white lines trace bathymetric contours at 200- and 400-meters depth,
while grey lines correspond to depths of 1000, 2000, 3000, 4000, and 4500 meters.

Accordingly, this research may be seen as a follow up investigation of the pioneering work by Thompson & Heywood
(2008), where the authors characterized comprehensively the WBCS of the Weddell Sea based on a high-resolution

hydrographic section counting with conductivity-temperature-depth (CTD) and lowered acoustic Doppler profiler (LADCP) measurements. This section was surveyed in 45,9 Sv (summer measurements) during a research cruise off the tip of the Antarctic Peninsula that crossed over the continental shelf and slope into the Weddell Sea following the above introduced
ADELIE line.

Ultimately, we designed this work to characterize the seasonal variability, in location and transport, of the multi-jet structure governing the western Weddell Sea from a climatological point of view based on two open-access reanalysis products, supporting our analyses with direct velocity measurements, altimetry data and a complete review of the state-of-the-art
knowledge in the literature. Furthermore, we also target to evaluate the goodness and convergence of these two open-access reanalysis products with different resolutions in delivering similar and coherent circulation patterns, both qualitatively and quantitatively.

The manuscript is organized as follows. A description of the methodology, reanalysis products, in situ and remotely sensed
observations are presented in section 2. Results and discussion are presented in section 3. Lastly, section 4 closes the manuscript with the conclusions.

## 2 Data and methods

Through this section we present the open-access numerical models that we analyse with an indication to their main configuration details. Next, we describe the set of Shipborne Acoustic Doppler Current profiler (SADCP) measurements
used to evaluate the agreement of the numerical model output data in capturing major oceanographic features recurrently reported in the literature. Lastly, we explain how we compute the volume transport to characterize the horizontal and vertical structure of the WBCS dynamically.

### 2.1 Data

In the following we present the details of the two global ocean circulation reanalysis products, Shipborne Acoustic Doppler
Current Profiler (SADCP) measurements and altimetry data that we use. Afterward, the calculations of the volume transport are addressed.

### 2.1.1 Reanalysis products

We use two global ocean circulation reanalysis products at different resolutions. These products are part of the Global Ocean Data Assimilation Experiment (GODAE), (Bell et al., 2009; Chassignet, 2011; Dombrowsky et al., 2009). A summary of
main details is presented in Table 1.



On the one hand, we use GLORYS2V4 (GLOBAL REANALYSIS PHY 001-031, https://doi.org/10.48670/moi-00024) and GLORYS12V1 (GLOBAL REANALYSIS PHY 001-030, https://doi.org/10.48670/moi-00021). They are global eddy-permitting and eddy-resolving ocean products of reanalysis, respectively, developed by the Copernicus Marine Environment

Monitoring Service (CMEMS). Thus, GLORYS2V4 and GLORYS12V1 data are provided daily with 0.25º and 0.08° of horizontal resolution, respectively. Bearing this in mind, at the regional scale of our high latitude study area, we must note that GLORYS2V4 data do not solve the mesoscale while GLORYS12V1 data does resolve the mesoscale as an eddy-solving ocean product. They are both based on the NEMO platform (Nucleus for European Modelling of the Ocean) and are atmospherically forced with ERA-Interim (https://www.ecmwf.int/en/forecasts/dataset/ecmwf-reanalysis-interim).


The water column in the former is divided in 75 while the latter is 50 vertical levels from the surface to the bottom. These reanalysis products are produced through assimilation of in situ and satellite data between 1993 and 2020. These databases deliver several ocean properties such as thermodynamic variables (temperature, salinity), dynamic variables (meridional and zonal total velocities, u and v, derived from sea surface height) as well as sea-ice related variables.


On the other hand, we use GLBv0.08 (expt_53.X, https://www.hycom.org/dataserver/gofs-3pt1/reanalysis), which is a version of the global HYCOM-based product (HYbrid Coordinate Ocean Model), supported by several institutions and sponsored by the National Ocean Partnership Program (NOPP). This product outputs are provided with 40 vertical levels, 0.08º of horizontal resolution between 40°S and 40°N, and 0.04º beyond these latitudes. This reanalysis product uses the

Navy Coupled Ocean Data Assimilation (NCODA) system and delivers output data with a temporal resolution of 3 hours after assimilating satellite and in situ sea surface observations, as well as vertical temperature and salinity profiles from XBTs, Argo floats and moored buoys. The numerical output data are available from 1994 to 2015.




| Model | GLORYS2V4 | GLORYS12V1 | GLBv0.08 - expt_53.X (GOFS 3.1) |
|---|---|---|---|
| **Ocean component** | ORCA025 LIM2 EVP NEMO 3.0, forced by ERA-Interim+ERA5, including SW+LW+PRECIP corrections<br><br>Eddy-permitting global ocean product<br><br>At high latitudes, this product does not solve the mesoscale | LIM2 EVP NEMO 3.1, forced by ECMWF ERA-Interim+ERA5, including SW+LW+PRECIP corrections<br><br>Eddy-resolving global ocean product<br><br>At high latitudes, this product also solves the mesoscale | HYCOM+NCODA, forced by NCEP CFS+NCEP CFSv2<br><br>Eddy-resolving global ocean product<br>At high latitudes, this product also solves the mesoscale |
| **Vertical mixing scheme** | Turbulent Kinetic Energy (TKE) with mixing length set to 10 m<br>Fixed coordinate | Turbulent Kinetic Energy (TKE) with mixing length set to 10 m<br>Fixed coordinate | f multiple vertical mixing turbulence closure schemes<br>Isopycnic default configuration |
| **Surface salinity restoring** | No global restoring strategy to sea surface salinity, but 3D-restoring towards EN4 products is applied below 2000m and poleward of 60°S (scale of 20 years) | Salinity information from top to bottom and 2D sea surface level | salinity with a 30-day relaxation time scale |
| **Sea-ice component** | Assimilation of sea ice concentration from CERSAT | Assimilation of sea ice concentration from CERSAT + OSISAF | Arctic Cap Nowcast/ Forecast System (ACNFS) |
| **Horizontal and vertical resolution** | 0.25° horizontal resolution and 75 vertical levels<br>Fixed coordinates | 0.08° horizontal resolution and 50 vertical levels<br>Fixed coordinates | 0.08° resolution between 40°S and 40°N, 0.04° poleward of these latitudes, 40 vertical levels<br>Lagrangian coordinates |
| **Bathymetry source** | ETOPO1 for deep ocean and GEBCO8 on coast and continental shelf | ETOPO1 for deep ocean and GEBCO8 on coast and continental shelf | NRL DBDB2 |

Table 1. Summary of the main configuration details for the reanalysis products of study.

**2.1.2 SADCP data**

We use data from two oceanographic cruises, pertaining to two different years and seasons. The initial dataset is derived from the R/V Laurence M. Gould cruise LG0003a, carried out in April 2000. The second dataset corresponds to the R/V Nathaniel B. Palmer, cruise NBP0106, conducted in November 2001. These datasets comprise direct velocity measurements obtained using 150 kHz 'narrow band' Shipboard Acoustic Doppler Current Profilers (SADCPs) along ship tracks, spanning



from the upper reference layer (~30 m) to approximately 500 m depth. Both datasets are openly accessible at: http://adcp.ucsd.edu/lmgould & http://currents.soest.hawaii.edu/nbpalmer.

230 **2.1.3. Altimetry data**

The surface geostrophic circulation of the study area is derived from SEALEVEL_GLO_PHY_L4_MY_008_047 (Global Ocean Gridded L4 Sea Surface Heights and Derived Variables Reprocessed 1993 Ongoing), hereafter ALT. This product is derived from various altimeter missions and encompasses data from GEOSAT to Jason-3. The altimeter data is processed using the DUACS multimission altimeter data processing system. The spatial resolution is 0.25° × 0.25°, with a temporal 235 resolution of daily covering the period from 1993 to 2020.

**2.2 Methods**

**2.2.1 The E-ADELIE transect**

To characterize the variability of the WBCS of the Weddell Gyre we focus on a key location, the historical ADELIE transect (Fig. 2), where the current system is well defined and has been previously described as a multi-jet structure current system 240 (Thompson & Heywood, 2008). The ADELIE transect is located northeast of the AP, where the Weddell waters may either turn around the Antarctic Peninsula towards the Bransfield Strait, leave the gyre circulation towards the Scotia Sea and the South Atlantic Ocean, or recirculate within the Weddell Gyre. Different from the traditional ADELIE transect, we analyse a version of that one which extends farther oceanward into the gyre interior and which we name E-ADELIE transect. This transect aims to account for the dynamics of the western branch of the Weddell Gyre to its full extent. We address the 245 comparison of the horizontal and vertical structure of the WBCS jets as depicted by the two reanalysis products, SADCP measurements and altimetry data.

**2.2.2 Volume transport**

For volume transport purposes, we compute the velocities perpendicular to E-ADELIE, namely v'. This procedure allows us to quantify the outflow of Weddell Sea waters across the E-ADELIE transect. The rotation angle is estimated from the 250 average angle of the E-ADELIE transect with respect to the true east. Thus, the Cartesian coordinates were rotated 15.79 degrees clockwise (α) from the true east following Eq. 1 (see the new reference frame in Fig. 1).

$$v' = v * \cos(\alpha) - u * \sin(\alpha) \qquad (1)$$

Subsequently, the rotated total volume transport (V'), hereafter referred as the cross-ADELIE volume transport, is estimated daily from v' at every time step of the complete time series of each product following Eq. 2:





$$V'(t) = \int_0^{D_L} \int_{-h}^0 v'(t) dz dx' \qquad (2)$$

where 0 and DL are the westernmost and easternmost limit of integration, respectively; and 0 and -h are the depths of integration from surface to bottom up to 4400m depth; and, v' is the time-dependent cross-ADELIE component of the velocity. Lastly, the time-averaged and associated standard deviation of V' is computed for every case of study.

### 2.2.3 Wind stress

We calculate wind stress as follows, using the formula proposed by Kara et al. (2013):


$$\tau = \rho \cdot U_{10}^2 \cdot C_D \qquad (3)$$

where $\rho$ represents the air density (1.2 kg m$^{-3}$); $U_{10} = \sqrt{u_{10}'^2 + v_{10}'^2}$ is the wind speed at 10 m above the surface (with $v_{10}$' and $v_{10}$' denoting rotated eastward and northward velocity components, respectively); and, $C_D$ is the drag coefficient, which

is a function of wind speed, $U_{10}$.

To ensure alignment with volume transport calculations, we compute cross- and along-ADELIE wind velocities. $v_{10}$' and $u_{10}$' are utilized to calculate $U_{10}$, maintaining the same rotation angle ($\alpha$) through the expressions:


$$v_{10}' = v_{10} * cos(\alpha) - u_{10} * sin(\alpha) \qquad (4)$$
$$u_{10}' = v_{10} * sin(\alpha) + u_{10} * cos(\alpha) \qquad (5)$$

We conduct basin-scale calculations to assess the average wind patterns across our study area. Subsequently, we analyse the relationship between the seasonal cycles of wind and volume transport, considering wind as a potential influencing factor.

To do this, we delineate the boundaries of the study are as follows: the northern boundary corresponds to a distinct change in wind direction at 64ºS; moving southward, the limit is established by identifying the point where the signal of the WBCS becomes evident, which occurs at 74ºS; the western boundary is demarcated by the Antarctic Peninsula (AP) at 62ºW, while the eastern boundary is determined by an observable shift in both wind and current directions at 38ºW.

The products employed for wind stress computation derived from the two reanalysis products used in this study. GLORYS2V4 and GLORYS12V1 use ERA-interim and ERA5 datasets, respectively. Since ERA-interim was discontinued in 2019, ERA5 forcing fields have been applied starting from January 2019, accessible at https://www.ecmwf.int/en/forecasts/dataset/ecmwf-reanalysis-interim and https://cds.climate.copernicus.eu/cdsapp#!/dataset/reanalysis-era5-single-levels?tab=overview. GLBv0.08 uses NCEP-CFS



and NCEP-CFSV2 products. NCEP-CFS was discontinued in 2010.        The corresponding data sets are accessible at https://www.hycom.org/dataserver/ncep-cfsr and https://www.hycom.org/dataserver/ncep-cfsv2.

## 3 Results and discussion

The applicability of Sverdrup dynamics for calculating the Weddell Gyre's transport is not straightforward and has been questioned for decades (Gordon et al. ,1981). In this latter work though, the authors estimated the Weddell Gyre transport in
~76 Sv, using wind stress data and applying Sverdrup balance. However, more recent studies using moorings and ship data have large ranges at lower estimates: 20-56 Sv in Fahrbach et al. (1991) and about 30 Sv in Yaremchuk et al. (1998). A handful of papers have addressed the Weddell Gyre's transport using idealized models, reanalysis products and observations. In this section, we assess the seasonal variations of the WBCS of the Weddell Sea gyre, as framed in the literature, in two open-access global ocean circulation reanalysis products at different resolutions (namely, GLORYS2V4, GLORYS12V1 and
GLBv0.08) and altimetry data. Also, we will use available SADCP observations collected in April 2000 and November 2001 to confirm the description of a new current.

We remind the reader that GLORYS2V4 data do not resolve the mesoscale at high latitudes while GLORYS12V1 data do as an eddy-resolving product. The primary difference between the NEMO-based products and the HYCOM-based product falls
on the ocean component of departure and the forcing, as indicated in Table 1. Also, the HYCOM-based product presents a notably higher resolution in our study area (0.04°) than GLORYS12V1 (0.08°).

The structure of the section is as follows. First, we analyse the horizontal structure of the surface velocity field of the WBCS of the Weddell Sea as represented in the two reanalysis products and observed from altimetry data (Fig. 3) and SADCP
observations (Fig. 4). Next, we analyse the vertical structure of the WBCS and volume transport variability based on the two reanalysis products (Fig. 5, 6 and Table 1-3), reviewing the estimated seasonality against the literature.

### 3.1 Horizontal structure

The spatial structure of the WBCS of the Weddell Gyre as observed at surface is presented in Fig. 3 (left-hand side panels) based on the time-averaged velocity field from altimetry (1993-2020), NEMO GLORYS2V4 (1993-2020), NEMO
GLORYS12V1 (1993-2020) and HYCOM GLBv0.08 (1994-2015), respectively (upper to lower panels).

The most prominent feature derived from altimetry data (geostrophic velocity field), and in all reanalysis products (total velocity) is the multi-jet structure of the WBCS, although less prominent in GLORYS2V4, and consisting of a series of parallel-aligned jets (CC, ASF, WF) running nearly perpendicular to the transect E-ADELIE, departing from the Antarctic
Peninsula's tip and oceanward (Figure 3). This multi-jet structure has been previously reported from both observational



works based on in situ observations (Muench & Gordon, 1995; Thompson & Heywood, 2008) and modelling studies (Stewart & Thompson, 2016; Matano et al., 2002).

In the real ocean, downstream E-ADELIE, one would find these jets diverging and splitting into different branches sourcing either Bransfield Strait, the Scotia Sea or recirculating within the gyre (Hellmer et al., 2005). One important note to this regard is the lack of a proper simulation of the branch feeding Bransfield Strait, which is absent in all reanalysis products but in altimetry data.

The identification of the jets as single features is more visible when looking at the velocity profile sampled along E-ADELIE, and shown in the right-hand side panels of Figure 3 (see the indication to seasons in the legend). Among datasets, we find several similarities and differences. The CC is absent as an off-shore and well-defined shelf jet in the altimeter-derived velocity field, likely due to land proximity in conflict with satellite measurements. Different to other data products, the CC in altimetry is captured as a coastal flow of increasing magnitude towards the shore. The ASF and the WF appear markedly at the positions where they are expected from the literature (Thompson and Heywood, 2008). Altimeter-derived peak speeds are also considerably lower when compared to SADCP & LADCP observations (Thompson and Heywood, 2008): CC (6 cm s-1 instead of ~20 cm s-1), ASF (4 cm s-1 instead of ~18 cm s-1) and WF (5 cm/s instead of ~12 cm s-1). Importantly, the altimetry climatologies indicate a confirmed recurrence through seasons (summer, autumn and spring) about the strength and position of all these jets. Lastly, we identify the presence of a broad, recurrent flow, less defined as a distinctive jet, but still persistent through seasons towards the interior of the gyre. When computing the volume transport associated with the WBCS of the Weddell Gyre, as will be shown and discussed later, we note this flow, about 150-200 km width, contributes significantly to the total volume transport. Present not only in altimetry data but also in all reanalysis products and direct velocity measurements from SADCP, we refer to this previously undescribed current hereafter as the Inner Weddell Current (IWC); flowing northeastward as a part of the inner western core of the Weddell Gyre, where interior waters recirculate around the centre of the gyre. We hypothesize this branch of the WBCS of the Weddell Gyre is mainly subject to the basin-scale wind-driven interior recirculation of the gyre and, subsequently, less influenced by topographic steering and thermohaline effects of the dense water formation along the continental slope.

When compared with GLORYS12V1, the multi-jet structure of the WBCS is less distinct in GLORYS2V4, indicating the limitations of the latter product's resolution (0.25°) to resolve the mesoscale in high latitudes and structures narrower than 25 km of spatial scale. As a result, the WBCS is revealed in GLORYS2V4 as a current system composed simply by: a coastal current (the CC); one rather broad, open-ocean current (~200 km width) embedding the ASF and WF; and, the IWC. This misrepresentation, or lack of details, of the multi-jet structure of the WBCS is expected after Chassignet (2011), who stated that at least 1/10° model resolution is needed to capture successfully the complexity of the western boundary current structures. In GLORYS12V1, the climatological CC and ASF appear as more distinctive jets, while the domain of the WF





and IWC is less defined and broadens over more than ~300 km. Also, the HYCOM-based product presents a notably higher resolution in our study area (0.04º), which may be the cause of some discrepancies when compared to GLORYS12V1 (Neme et al., 2021; Renner et al., 2009). To this regard, Renner et al. (2009) emphasizes that increasing the model resolution is not always the best choice, as eddy-permitting and eddy-resolving products are usually better parametrized and increasing the resolution further may require a reformulation of those parameterizations.


Regarding Fig. 4, we must note that: (1) the SADCP transect is not exactly aligned with E-ADELIE in the reanalysis products; and, (2) the SADCP observations correspond to a synoptic transect while the reanalysis products-based data correspond to climatological transects. However, we find the SADCP transects serves well to the purpose of comparing major features across the WBCS of the Weddell Sea.


Noting the different x-axis for panels b-d in Fig. 4, and comparing the locations of these SADCP observational transects and E-ADELIE in Fig. 2, we find that an agreement exists between the relative location of the Antarctic Coastal Current, Antarctic Slope Front and Weddell Front in all velocity profiles in Fig.3 but for GLORYS4V2 in panel d, where the ASF and the WF are not presented as separate jets. Thus, the locations of the jets associated with the CC, ASF and WF along the

SADCP transects, although appear at different distances than those displayed along E-ADELIE for GLORYS12V1 and GLBv0.08, agree well when projecting them onto E-ADELIE (see these distance projections in Fig. 2-4). These locations also agree with those found along the ADELIE transect discussed in Thompson & Heywood (2008) in their Fig. 9.

Hence, in line with the SADCP observations we find that, along E-EDALIE, GLORYS12V1 and GLBv0.08 present the CC

within the first 100 km, the ASF centred at around 150 km and, lastly, the WF centred at around 260 km with a width approaching 100 km in all cases (Fig. 4). As one could expect, the strength of the jets is higher in the synoptic SADCP observations compared to the climatological values of the reanalysis data. However, there are large differences (up to one order of magnitude higher) for speed values of the CC when comparing the values of the climatological reanalysis products (4-8 cm/s) against the synoptic SADCP observations (30-90 cm/s). This discrepancy may be attributed to bathymetric

influences within the CC region, with the SADCP transects traversing depths ranging from 400m to 200m and back to 400m. The jets associated with the ASF and WF also display synoptic speed values up to one order of magnitude larger than modelled speed values, although they are generally much weaker than the CC (6-8 cm/s for the reanalysis products data and up to 40 cm/s in the SADCP observations).

With regards to seasonality, GLORYS4V2 and GLORYS12V1 display no significant variability in either strength or spatial location while GLBv0.08 does. In the latter, the signal of the CC is absent during summer and strongest, and about 9 cm/s, during winter while the jets associated with the ASF and WF are strongest during spring with mean values being up to 25% higher compared to the weakest seasons (summer and autumn). The IWC is observed between 300-350 km and 600 km



offshore the Antarctic Peninsula in altimetry data and all the reanalysis products but GLBv0.08 (Fig. 3), ranging roughly

between 2-8 cm/s and peaking at 40 cm/s in the SADCP observations (Fig. 4). Once again, the highest seasonal variability is found for GLBv0.08 with the IWC being strongest at 10 cm/s during summer.

The weaker strength of these currents and jets in the modelled data as compared to observations might be attributed to two factors. On the one hand, the different time-scales involved in the comparison: the SADCP data are synoptic measurements

and the modelled data are climatological values obtained after a time-averaging process (always expected to be smoother values). On the other hand, the three products assimilate remotely-sensed observations, including scatterometer and altimetry data, which help the products to adjust numerical solutions to measured surface ocean currents. If we account that remotely-sensed derived surface currents (geostrophic plus Ekman currents) have been reported to underestimate direct velocity measurements, for instance by 27% on average in the Agulhas Current System but ranging by 4-64% (Hart-Davis et al.,

2018), one can reasonably understand the magnitude offset in the reanalysis product data. Furthermore, regions rich in high mesoscale variability such as eddies, current meandering and instability waves are expected to result in simulation errors, especially when major currents are not represented by strong flows (Hewitt et al., 2020).






Figure 3. Panels a, c, e, and g present the horizontal structure of the WBCS of the Weddell Gyre following the time-averaged velocity field (cm/s) at surface. The displayed arrows represent unitary vectors, and E-ADELIE is indicated with a black line. Panels b, d, f, h depict the seasonal surface velocity field (cm/s) along E-ADELIE with its standard deviation. In panel (b), the winter season is absent, based on the requirement that at least more than 50% of the dataset along the E-ADELIE contains not empty values. Computations encompass the
complete time series of each dataset for all panels.




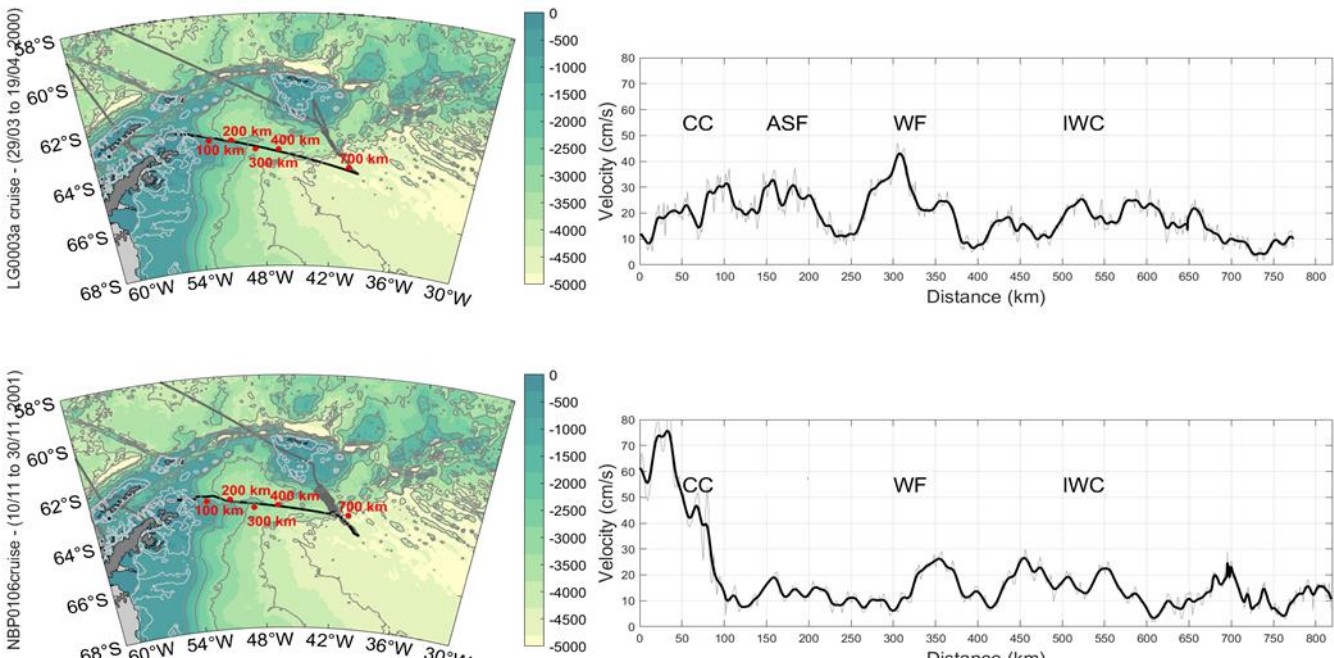

Figure 4. Panels a and c show the bathymetry of the north-western sector of the Weddell Gyre depicting the area studied during two oceanographic campaigns in different seasons and years, corresponding to: (a) LG0003a of R/V Laurence M. Gould in April 2000, and (c) NBP0106 cruise of R/V Nathaniel B. Palmer in November 2001. The transects are shown in gross grey lines, and the area studied on this work is highlighted in black. Red dots indicate the distance along the transects. Panels b and d picture the corresponding velocity field (cm/s) of the cruises on the left, coming from SADCP measurements in the domain of E-ADELIE. Depth-averaged SADCP measurements (25-55 m) are presented as original data in grey, smoothed data in black. We do not indicate the ASF in panel (d) as a closer inspection to the current direction indicates a reversed flow over this location.

## 3.2 Vertical structure and cumulative volume transport

The vertical structure of the WBCS through time-averaged vertical sections of the velocity field (v') and the cumulative transport (V') along E-ADELIE are presented in Fig. 5 for GLORYS2V4, GLORYS12V1 and GLBv0.08, investigating both depth and width of each branch. See in Section 2.2.2 how the velocity field has been rotated.

Fig. 5 reveals the multi-jet structure and barotropic nature of the WBCS jets. This multi-jet structure is absent in GLORYS2V4 and mostly visible in GLORYS12V1 and GLBv0.08 (Fig. 5a, c, e), as anticipated in Fig. 3. GLORYS2V4 displays a broad current flowing northward (positive values), particularly intensified between 125 km and 450 km offshore, while GLORYS12V1 and GLBv0.08 capture a more complex structure featuring three distinct jets (CC, ASF, WF) and a



broader current (IWC) towards the interior of the gyre, as previously noticed from the surface view in Fig. 3 (altimetry and reanalysis data) and Fig. 4 (SADCP measurements).

The finer resolution of GLORYS12V1 and GLBv0.08 allows for a more detailed depiction of the jets at depth. Thus, the CC,
ASF and WF appear centred at offshore distances from the eastern Antarctic Peninsula coastline around 50, 160, and 240 km, respectively, as bottom-intensified equivalent barotropic jets. These jet locations are in agreement with previous works (Thompson & Heywood, 2008; Kerr et al., 2012; Stewart & Thompson, 2016; Thompson et al., 2018; Vernet et al., 2019). Additionally, we find the IWC appears farther offshore from 250 km towards 600 km in GLORYS12V1 and GLBv0.08. Within the broad domain of the IWC, two well-defined jets are prominent at offshore distances about 370 km and 420 km in
both GLORYS12V1 and GLBv0.08. However, the strength of the IWC decreases sharply below 2 cm/s at offshore distances about ~470 km in GLORYS12V1 and farther offshore, at about 600 km, in GLBv0.08.

As for the vertical resolution, GLORYS2V4, GLORYS12V1 and GLBv0.08 cover the upper 2000 m of the water column with ~40 depth levels. Below this depth, the resolution in all the products decreases sharply, being 9 depth levels in
GLORYS2V4, GLORYS12V1 and only 4 depth levels GLBv0.08 product. To this regard, we must note that the vertical scheme might be a key factor since vertical coordinates in HYCOM remain isopycnic in the open, stratified ocean and smoothly transition to z coordinates in the weakly-stratified upper-ocean mixed layer while both GLORYS products are fixed coordinate. To this regard, the isopycnic configuration in HYCOM has proved to misrepresent the bottom boundary layers, because they are notably unstratified (Chassignet, 2011).


Finally, the oceanward distribution of the cumulative transport, V', within E-ADELIE displays a common pattern where transport values increase markedly between a distance of 150 km and 470 km offshore in GLORYS2V4 and GLORYS12V1, and between 150 km and 600 km offshore in GLBv0.08. This increase is nearly linear in GLORYS2V4 and GLORYS12V1 and about 0.075 Sv per km and 0.076 Sv per km, respectively. Differently, in GLBv0.08 we distinguish two segments of
different slopes for this transport increase. The first one is found to be between 150 km and 350 km and about 0.064 Sv per km; the second one is found to be between 350 km and 600 km and about 0.156 Sv per km. If we perform a similar analysis to find the rate of volume transport increase by distance but now based on the direct velocity measurements reported in Thompson and Heywood (2008), our calculations yield a rate of 0.11 Sv per km (derived from transport estimates in their Fig. 11). This rate of volume transport increase approaches our estimates based on reanalysis products. Following the
cumulative transport shown in Thompson and Heywood (2008) along the ADELIE transect, their Fig. 11, the authors report approximately 46 Sv. Our estimates for a similar offshore transect length about 400 km lead to relatively lower transport estimates: 24.41 Sv for GLORYS2V4, 23.45 Sv for GLORYS12V1, and 19.38 Sv for GLBv0.08. We attribute these lower values in the reanalysis products, among other factors previously discussed, to their climatological nature compared to the synoptic LADCP measurements in Thompson and Heywood (2008).






Beyond the above offshore distances for linear volume transport increases (offshore distances of 470 km in GLORYS2V4 and GLORYS12V1, and 600 km in GLBv0.08), the cumulative transport remains nearly constant and about 36 Sv in GLORYS2V4 and GLORYS12V1, and about 56 Sv in GLBv0.08. The above description highlights that GLBv0.08 represents the WBCS of the Weddell Sea Gyre as a stronger and wider multi-jet system, extending a significantly important

branch of transport recirculation about 130 km farther to the east than in GLORYS2V4 and GLORYS12V1.

With regards to that interior branch, the IWC, we obtain cross-transect surface (0-300 m depth) volume transports about 6.67 Sv and 7.64 Sv from the SADCP measurements taken along the LG0003a and NBP0106 surveys, respectively. When computing the same estimates from the reanalysis products, we obtain lower volume transports values ranging from 2-4 Sv.

If we do the same calculations from the reanalysis products but now from the surface down to 4400 m depth, and from 50 km east of the Weddell Front up to 600 Km offshore from the Antarctic Peninsula, cross-transect volume transports reach about ~19±4 Sv in GLORYS2V4, 21±3 Sv in GLORYS12V1, and ~43±4 Sv in HYCOM.



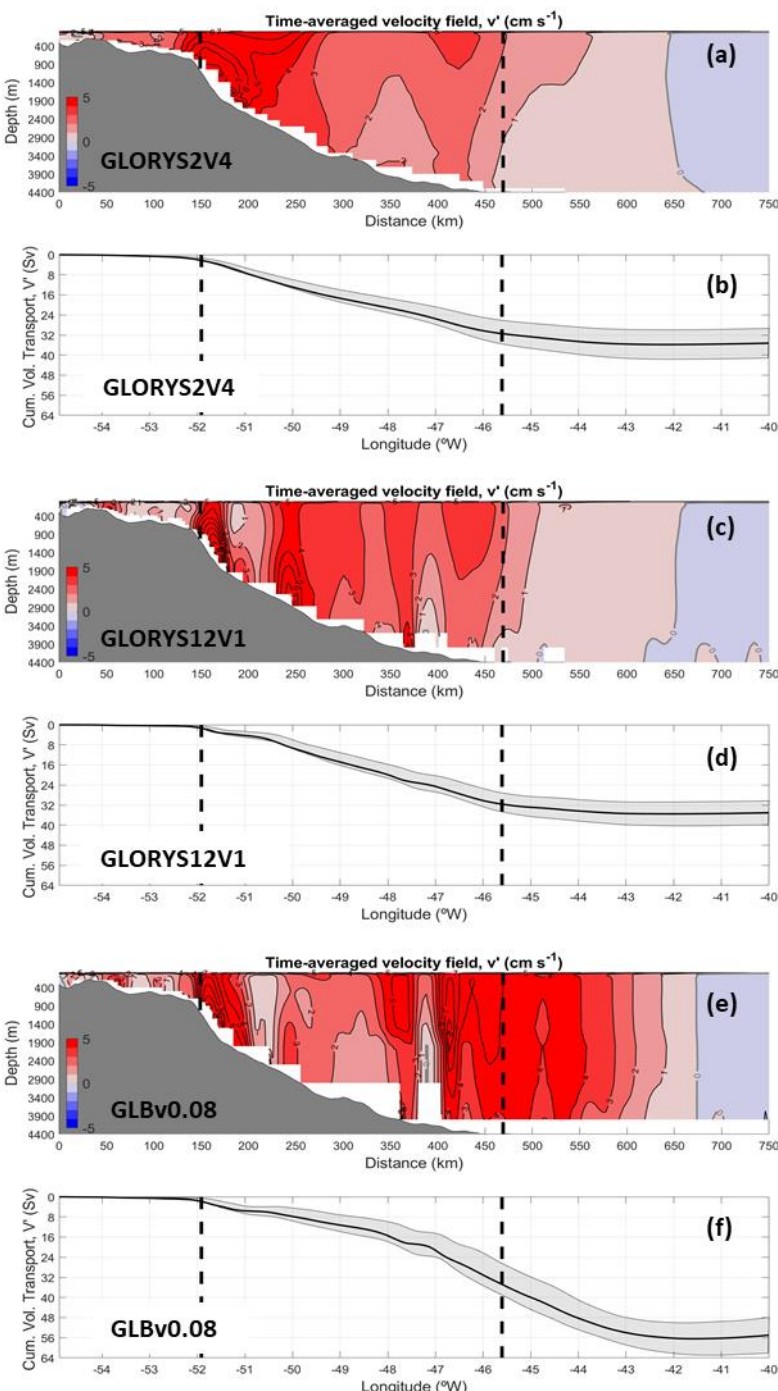

Figure 5. (a, c, e) Time-averaged velocity field of v' (cm/s) for E-ADELIE as modelled in GLORYS2V4, GLORYS12V1 and GLBv0.08, respectively. (b, d, f) Cumulative transport, V' (Sv) of the time-averaged velocity field of v' for E-ADELIE as modelled in GLORYS2V4,



GLORYS12V1 and GLBv0.08, respectively. The time-average corresponds to the available coverage of each reanalysis product. Dashed black lines are used as reference for discussion of the results in the text.

### 3.3 Seasonal variations of the volume transport

We assess the seasonal variations across E-ADELIE for GLORYS2V4, GLORYS12V1 and GLBv0.08, based on bottom-reaching volume transport estimates following Equation (2), where the length of the transect for integration, DL, extends between offshore distances of 150 km and 470 km. This choice responds to the area embedding the WBCS major transports in GLORYS2V4 and GLORYS12V1. Additionally, we also compute volume transport estimates where DL extends farther offshore between distances of 150 km and 600 km but only for GLBv0.08, given its broader high-transport spatial domain

along the WBCS of the Weddell Gyre. We must note that beyond 450 km offshore the seafloor is deeper than 4400 m, but we set this as the bottom boundary for integration between 450 km and 600 km. Results are presented in Fig. 6 and summarized in Table 2.

Table 2 highlights that GLORYS2V4, GLORYS12V1 and GLBv0.08 provide similar time-averaged estimates of volume

transport, V', about 30 Sv. Also, GLORYS12V1 displays the lowest standard deviation (~3.83 Sv) as opposed to GLBv0.08, which displays the highest (~5.88 Sv). When comparing the estimates for the total volume transport and estimates accounting only for the northward volume transport (positive values), we find differences lower than 1 Sv in the three products, which implies that the southward flow in this region is almost negligible by comparison.  As one could reasonably expected, the time-averaged estimates for GLBv0.08* almost doubles the time-averaged estimates for the shorter section of

E-ADELIE.

In Fig. 6, the three panels display a clear seasonal cycle with maximum values through autumn-to-winter and minimum values through summer-to-spring, encompassing cross-transect volume transport, V' (panel a), cross-transect surface (0-100m) volume transport, V' (panel b), and basin-scale wind-stress (panel c). This pattern agrees well with the modelled

seasonal cycle of volume transport and wind stress curl for a shortened version of the ADELIE transect focused on the Weddell Sea Bottom Water transport (Wang et al., 2012).

In Fig. 6a, the seasonal cycles from GLORYS2V4 and GLORYS12V1 follow each other closely, being always about 2 Sv higher in GLORYS12V1 through the months transitioning to/from maxima values. Both products agree on peak volume

transports reaching 32-33 Sv in autumn and winter months. The minima appear in January at 24 Sv and 26 Sv, respectively, for GLORYS2V4 and GLORYS12V1. This leads to a seasonal amplitude of about 7-8 Sv. In GLBv0.08, volume transport estimates computed for the same transect length (DL between 150 km and 470 km offshore), lead to a more conspicuous seasonal cycle with a maximum mean transport of 37 Sv in July (winter) and minimum mean transport of 21.5 Sv in December (summer), leading to a larger seasonal amplitude about 15.5 Sv, as compared to GLORYS products. This larger



seasonal amplitude occurs because GLBv0.08 seems to overestimate (underestimates) the seasons with highest (lowest) transports. When computing the volume transport in GLBv0.08 for the longer section (DL between 150 km and 600 km and accounting for the ASF, WF and broader IWC), we refer to this as GLBv0.08*. Then, the maximum and minimum transports occur through the same months as in GLBv0.08 but now reach 56 Sv and 48 Sv, respectively (decreasing its seasonal amplitude of variation down to 8 Sv). The latter suggests that most of the WBCS seasonal variability in GLBv0.08 is

embedded between 150 km and 470 km distances offshore the Antarctic Peninsula.

In Fig. 6b (cross-transect surface volume transport), GLORYS2V4 and GLORYS12V1 align well in capturing the months of peak transports, but exhibit relatively greater discrepancies compared to panel a (about 27-44% of the total estimates). Notably, the disparity between the two products reaches 0.4 Sv (winter values reach almost 1.5 Sv in GLORYS2V4 and 1.1

Sv in GLORYS12V1; conversely, for summer values, we observe 1.1 Sv in GLORYS2V4 and 0.9 Sv in GLORYS12V1). Generally, GLORYS2V4 and GLBv0.08 follow each other more closely regarding surface V' estimates.

In Fig. 6c, upon comparing seasonality of V', and surface V', with seasonality of the wind stress spatially averaged over the study area using ERA5, Era-Interim and NCEP CFS as wind data sources, it becomes evident that the three panels

predominantly portray the same pattern. This observation prompted us to investigate the significance of the wind stress as the major driver for the meridional transport in this region. Accordingly, we explored the correlation between the cross-transect volume transports in panels a and b with the wind stress in panel c, revealing high and significant correlations among the different products (Table 3), particularly noteworthy for GLORYS12V1. The correlation estimates for GLBv0.08, despite being high, at least in the case of the cross-transect surface V', were lower compared to those derived from both

GLORYS datasets. To this regard, we find worth recalling the different atmospheric forcing for each product, ERA5, ERA-Interim for the NEMO-based products (GLORYS2V4, 1.1 Sv in GLORYS12V1) and NCEP for the HYCOM-based product (GLBv0.08) (Table 1).

The above results are in agreement with the theoretical framework of a wind-driven gyre forced by the basin-scale seasonal

variability of the wind-stress curl (Azaneu et al., 2017; Franco et al., 2007; Gyldenfeldt et al., 2002; Le Paih et al., 2010; Wang et al., 2012). In this context, we recall the reader that the average wind-forcing acting over the basin is featured by the westerlies at the lowest latitudes of the gyre, and easterly winds near the coast. This forcing has a strong seasonality with weakened winds during the austral summer, when the wind-driven currents are reduced by ~50% as compared to values during winter and fall (Armitage et al., 2018; Talley et al., 2011; Schröder & Fahrbach, 1999). Accordingly, all products

reveal a coherent cycle for the volume transport driven by the WBCS of the Weddell Gyre across E-ADELIE, with maximum transport values occurring during autumn-to-winter and minimum values during summer-to-spring in agreement with former modelling efforts in the literature (Wang et al., 2012).



Lastly, we have included in Fig. 6 a set of in situ-based estimates of volume transport as reported in the literature (values listed in Table 4). These estimates are represented by dots when they refer to synoptic measurements and by dashed-lines

when they refer to time-averaged values. Generally, in situ-based estimates are notably closer to modelled-based estimates obtained with GLORYS2V4, GLORYS12V1 and GLBv0.08 for DL between offshore distances of 150 km and 470 km. It is also worth mentioning that the in situ-based estimates are full-depth integrated volume transports computed across transects analogous to E-ADELIE. This agreement is primarily observed during the summer and early autumn periods due to the challenging weather conditions prevalent in the study area during other seasons. Regarding the time-averaged values based

on observations, as outlined in Table 4, it becomes apparent that only the estimates provided by Matano et al. (2002) and Kerr et al. (2012) deviate from the climatological seasonal estimates computed from the reanalysis products, displaying values well below the established averages. When compared to GLBv0.08*, the latter presents transport values which are typically higher than those reported in the literature for ADELIE. Unfortunately, the absence of volume transport estimates based on observations or modelling data for the longer section computed with GLBv0.08*, prevent us from discussing

further these results. We note that access to analogous observational-based or reanalysis product-based approaches are required in order to assess these results robustly. However, despite the lack of analogous observational data, we hypothesise that the difference between NEMO-based and HYCOM-based products over the IWC area, may rely on the fact that the Weddell Gyre is extremely sensitive to horizontal resolution, as reported in Styles et al. (2023).

Generally, we find the above comparisons with observational-based and reanalysis product-based estimates in the literature support the products' performance, particularly in the case of GLORYS2V4, GLORYS12V1, and GLBv0.08 for DL between offshore distances of 150 km and 470 km.

| Model | Total volume transport of V' (1) in Sv | Positive volume transport of V' (2) in Sv | Section |
|---|---|---|---|
| GLORYS2V4 | 29.29 ± 4.54 | 29.31 ± 4.54 | E-ADELIE enclosed between 150 and 470 km offshore |
| GLORYS12V1 | 30.65 ± 3.83 | 30.83 ± 3.74 | |
| GLBv0.08 | 31.80 ± 5.88 | 32.31 ± 5.59 | |
| GLBv0.08* | 52.25 ± 5.13 | 52.95 ± 4.74 | E-ADELIE enclosed between 150 and 600 km offshore |

Table 2. Time-averaged seasonal volume transports across E-ADELIE for the three products of study between 150 km and 470 km offshore the Antarctic Peninsula. The numbers (1) and (2) as superscripts indicate the following: (1) indicates that both northward and southward flows are considered to compute the volume transport; (2) indicates that only northward flows (positive values) are considered. GLBv0.08* indicates that transport estimates are computed between 150 km and 600 km offshore the Antarctic Peninsula.





| Reanalysis product | $R^2$ between surface V' and wind-stress | $R^2$ between total V' and wind-stress |
|---|---|---|
| GLORYS2V4 | 0.799 | 0.670 |
| GLORYS12V1 | 0.848 | 0.692 |
| GLBv0.08 | 0.686 | 0.362 |


Table 3. Results derived from linear model regression, where y represents the total and surface cross-transect volume transports in each scenario, and x denotes the wind stress. All reported values exhibit p-values approaching 0 and are presented with 95% coefficient bounds.







Figure 6. (a) Monthly climatology of the volume transport, V', across E-ADELIE computed between 150 km and 470 km offshore the Antarctic Peninsula with their corresponding standard deviation (shades around the climatological values). The coloured thick lines (see legend) refer to each product as follows: grey solid line (GLORYS2V4), black solid line (GLORYS12V) and magenta solid line (GLBv0.08). GLBv0.08* in red solid line (as indicated in the legend) refers to the monthly climatology of the volume transport, V', across E-ADELIE for a longer transect extent (DL between 150 km and 600 km offshore the Antarctic Peninsula). Observational data for

analogous transects to E-ADELIE reported in the literature, and listed in Table 4, are plotted in coloured dots when corresponding to measurements taken at a synoptic scale, and in horizontal dashed lines when corresponding to a time-averaged estimate. (b) Same as panel above, but computed for the surface layer (from 0 to 100 m depth). (c) Wind stress seasonal cycle in Pa, computed over the study region.

| Reference | Data Source | Current | Section Name | Time | Depths of integration | Volume transport (Sv) |
|---|---|---|---|---|---|---|
| **Fahrbach _et al._ (1994)** | Current meter | Western Boundary Current System | SR04 WOCE 500 km | 1989 to 1991 | 0-4000 m | 29.5 ± 9.5 |
| **Muench & Gordon (1995) \*** | Current meter | Western Boundary Current System | Transect 3, 68ºS - 200 km | February-June, 1992 | 0-4000 m | 28 |
| **Garabato _et al._ (2002) \*** | LADCP | Northward South Orkney Plateau | ALBATROSS 1400 km | March-April, 1995 | 200-4000 m | 27 ± 7 |
| **Matano _et al._ (2002)** | Modular Ocean Model, Vr2 | Western Boundary Current System | 47-55ºW, 65ºS | 1979 to 1998 | 500-3000 m | 18 |
| **Thompson & Heywood (2008) \*** | LADCP | Coastal Current | ADELIE 300 km | February, 2007 | 0-4000 m | 1.3 |
| | | Slope front | | | 0-4000 m | 3.9 |
| | | Weddell front | | | 0-4000 m | 16.8 |
| **Kerr _et al._ (2012)** | OCCAM global model (1/12º) | Western Boundary Current System | 350 km | 20-year averaged simulation | 500-4000 m | 28.5 ± 2.9 |
| **Palmer _et al._ (2012) \*** | Inverse model - ADCP | Northward Powell Basin | 600 km | January, 2008 | 0-5000 m | 25 |
| **Wang _et al._ (2012)** | FESOM model | Tip of the Antarctic Peninsula | Transect D1-D2 | 2001 to 2011 | 0-5000 m | 20 ~ 24 |



| Jullion *et al.* (2014) * | Inverse model box | Southward South Scotia Ridge | Merged transect ANDREX/I6S | January, 2009 | 0-5000 m | 24 ± 4 |
|---|---|---|---|---|---|---|
| Reeve *et al.* (2019) | Argo float | Western Boundary Current System | Section 6 | 2002 to 2016 | 0-5000 m | 27 ± 5 |

Table 4. Review of volume transport estimates computed across transects analogous to ADELIE and based on both observational and model data. The symbol * refers to single observations, while its absence indicates time-averaged estimates over the period of time indicated in the table.

## 4 Conclusions

We have conducted an analysis between two open-access global ocean circulation reanalysis products based on NEMO and HYCOM configurations and operating at distinct resolutions to investigate the seasonal variations of the Western Boundary Current System of the Weddell Sea Gyre. The reanalysis products are known as: GLORYS2V4 (global ocean eddy-permitting product), GLORYS12V1 (global ocean eddy-resolving product), and GLBv0.08 (global ocean eddy-resolving product). We support the analysis further with in situ direct velocity measurements along the transect of study, altimetry data and discussions against prior studies.

We focus on an extended version of the historical ADELIE transect, which expands oceanward from the northernmost tip of the Antarctic Peninsula into the interior of the gyre. This extension is referred to as the Extended ADELIE transect (E-ADELIE). Understanding the dynamics of the WBCS across E-ADELIE is crucial as it serves as a critical gateway where water masses either leave the basin or circulate within the gyre again, potentially causing downstream effects on the thermohaline circulation. Through the investigation of the horizontal and vertical structure of the WBCS, as well as its seasonal variability in volume transport, the following findings have been achieved.

When considering the horizontal structure, the multi-jet configuration of the WBCS has been consistently observed across all products, except for GLORYS2V4. The presence of parallel-aligned jets along the slope of the Antarctic Peninsula has been well-established through both observational studies and modelling efforts: the Antarctic Coastal Current, Antarctic Slope Front, and Weddell Front. Notably, the resolution limitations of GLORYS2V4 have led to a less distinct multi-jet structure, highlighting the importance of high-resolution products for capturing the complexity of Western Boundary Current systems in high latitudes. The lack of this feature has been attributed to constrained mesoscale resolution at high latitudes.



The vertical structure analysis reveals that GLORYS12V1 and GLBv0.08, due to their finer resolutions, also offer a more detailed depiction of the multi-jet structure at depth, with distinct bottom-intensified jets in agreement with observations. Furthermore, the relative location of the CC, ASF and WF in the reanalysis products aligns well with their location as seen from in situ direct velocity measurements, altimetry data and previous studies. Despite discrepancies about the strength and horizontal extent, the three reanalysis products, altimetry and SADCP data indicate the presence of a previously unreported current which seems to drive largely the recirculation of interior waters within the gyre. We name this novel feature as the Inner Weddell Current.

In the context of cumulative volume transport, consistent trends per distance in agreement with observations have been observed across GLORYS2V4 and GLORYS12V1 running along E-ADELIE. In GLBv0.08, the wider and stronger presence of the Inner Weddell Current deviates its cumulative volume transport towards larger estimates when compared to the NEMO-based products and with previous studies along E-ADELIE.

Remarkably, the analysis of seasonal variations in volume transport aligns well with the expected impact of the wind stress variations, supported with strong and significant correlations. These estimates show that GLORYS2V4, GLORYS12V1 and GLBv0.08 also appear to respond largely to the basin-scale wind stress forcing as demonstrated in previous modelling studies using different models. Thus, the seasonal cycle for the cross-transect volume transport across E-ADELIE is characterized by common minimum values around $25 \pm 5$ Sv during the period from September to December, and peak values of approximately $33 \pm 5$ Sv between March and July. Correspondingly, the time-averaged transport levels converged around $30 \pm 5$ Sv across all products. Because GLBv0.08 revealed an expanded and stronger Inner Weddell Current (~150 km) to the east of the Weddell Front, volume transport estimates were also computed accounting for a longer section, then delivering an average volume transport of about $52 \pm 5$ Sv.

These analyses highlight the challenge of reconciling modelled data with observations due to differences in spatio-temporal scales and data sources where weather conditions make year-round, full-depth in situ measurements inaccessible. As our understanding of the Weddell Sea gyre continues to evolve, the insights gained from this study contribute to refining ocean products and enhancing our comprehension of WBCS in high latitudes. We expect our findings to set a baseline of groundwork about the seasonal dynamics of the WBCS of the Weddell Sea regarding its horizontal, vertical structure and associated volume transport and to open an avenue for future analysis in a key area for global ocean circulation where limitations to access in situ observational data exist.



**Code availability**

The software codes utilized in this study are accessible upon request from the primary author. However, the specific code utilized for wind data processing can be obtained from the following source: https://www.chadagreene.com/CDT/windstress_documentation.html#9 (Kara et al., 2013).

**Data availability**

All data we used in this investigation are publicly accessible on the web pages referenced throughout the text (section 2.1).

**Author contributions**

In accordance with the CRediT contributor roles taxonomy, the author contributions can be outlined as follows: **Tania Pereira-Vázquez**: Conceptualization, Data Curation, Formal Analysis, Methodology, Validation, Visualization, Writing -
original draft; **Borja Aguiar-González**: Conceptualization, Investigation, Methodology, Supervision, Writing - original draft; **Ángeles Marrero-Díaz**: Conceptualization, Data Curation, Investigation, Funding acquisition, Supervision, Writing - review & editing; **Marta Veny**: Conceptualization, Formal Analysis, Visualization, Writing - review & editing; **Ángel Rodríguez- Santana**: Conceptualization, Investigation, Funding acquisition, Writing - review & editing.

**Declaration of Competing Interest**

The authors declare that they have no known competing financial interests or personal relationships that could have appeared to influence the work reported in this paper.

**Acknowledgements**

This work has been supported by the EUROPEAN COMMISSION-Research Executive Agency (REA) through the project MISSION ATLANTIC (Grant agreement ID: 862428), owing to the program/call H2020-BG-2018-2020 / H2020-BG-2019-
2. This work has also been co-financed by the Agencia Canaria de Investigación, Innovación y Sociedad de la Información de la Consejería de Universidades, Ciencia e Innovación y Cultura, and by the European Social Fund Plus (ESF+) Integrated Operational Program of the Canary Islands 2021-2027, Axis 3 Priority Theme 74 (85%). The first author acknowledges the financial support provided by these institutions through a PhD scholarship (TESIS2022010091). ChatGPT, developed by OpenAI, was used for proofreading of this manuscript.



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

**Appendix**

Figure A1 summarizes the hydrography of the Western Boundary Current System of the Weddell Gyre as introduced in section 1 of this study. Thus, we present vertical sections of potential temperature and salinity for the ADELIE transect based on in situ measurements from the Cruise SOS-Climate II of RV Ary Rongel in February 2009 and modelling data (GLORYS2V4, GLORYS12V1, GLBv0.08).

Generally speaking, a good agreement exists among the four views, where major water masses described in section 1 are
present at the characteristic horizontal and vertical scales, as well as within the characteristic temperature-salinity ranges. It is important to note that none of the three reanalysis products assimilated these observational data. One exception occurs with the Weddell Sea Bottom Water, which is absent in GLORYS2V4 and not accurately modelled in GLBv0.08.





Figure A1. Vertical sections of potential temperature and salinity along the ADELIE transect. Panels (a) and (b) depict in situ measurements during Cruise SOS-Climate II of RV Ary Rongel in February 2009. Panels (c) to (h) showcase the reanalysis products



employed in this study. The figure includes the key water masses for this region: Antarctic Surface Water (AASW), Warm Deep Water (WDW), and Weddell Sea Bottom Water (WSBW).
