# Peer review of "On the Seasonal Western Boundary Current System of the Weddell Gyre"

_EGUsphere, 2024_

## Referee Comment (RC1)

[referee-annotated manuscript omitted]

---

## Community Comment (CC3)

Dear Reviewer #1,

We appreciate the time and efforts you have dedicated in providing feedback about our manuscript and are grateful for the insightful comments that have contributed to improve our research work. We have now finalized the revision of our manuscript entitled "On the Seasonal Western Boundary Current System of the Weddell Gyre" for final consideration to the Journal of Ocean Science.

In the following, a detailed point-by-point response to your comments is presented. To make a clear distinction, comments from the reviewer #1 are marked in bold font while our response is in regular font. To ease their identification through this document, new text for the revised version of the manuscript is highlighted in blue font.

Please note that the lines indicated as LXXX in our response refer in all cases to the submitted manuscript. Please, note that new figures and tables mentioned in our response are presented at the end of this document. These new figures and tables will be indicated as Figure RY or Table RY, where R stands for Revision and Y stands for an ascending number. If we refer to Figures and Tables of the submitted manuscript, we will use the same format as in the submitted manuscript.

Before proceeding with the detailed point-by-point response, we provide below an overview of the main changes applied in the revised version to ease its assessment:

1.- After performing the analyses suggested by the reviewers, and carefully addressing their concerns jointly, we find their suggestions add so many important aspects to the story that the space left for discussion of differences among NEMO and HYCOM led us to decide to drop the latter from the revised version. We are happy the first submitted version shows the major differences among products, but we agree that maybe we were trying to cover too much at once and by doing so the main message may be unclear. The revised paper focuses on NEMO-based products and observations (direct velocity measurements and altimetry data) solely. We use this variety of data to characterize the seasonal variations of the Western Boundary Current System of the Weddell Gyre assessing an open-access product against existing observations and former modeling studies in order to enable its future use in studies about the interannual variability of this current system. Also, we describe the ocean dynamics governing the interior branch of the ocean gyre, demonstrating this is highly controlled by the basin-wide wind stress forcing as opposed to the most relevant role that thermohaline forcing plays as one approaches the coastal zone (sea-ice formation/melting dominates here).

*Following the suggestion by #Reviewer1: to drop the HYCOM-based results from the manuscript.*

2.- We have recomputed the wind stress-forcing in all cases accounting for the presence of sea-ice making use of the algorithms develop by Greene et al., 2019:

Chad A. Greene, Kaustubh Thirumalai, Kelly A. Kearney, José Miguel Delgado, Wolfgang Schwanghart, Natalie S. Wolfenbarger, Kristen M. Thyng, David E. Gwyther, Alex S. Gardner, and Donald D. Blankenship. The Climate Data Toolbox for MATLAB. Geochemistry, Geophysics, Geosystems 2019. doi:10.1029/2019GC008392 https://doi.org/10.1029/2019GC008392.

*Following the suggestion by #Reviewer2: to account for the sea-ice presence when computing wind stress.*

Also, we have included in our analysis correlations of the volume transport and the wind stress curl.

*Following the suggestion by #Reviewer1: to account for the basin-wide wind forcing acting over the ocean gyre.*

3.- We have filled in the gaps near the bottom-sea accounting for the bottom-intensified jets. To do so we have used an algorithm available in MATLAB code that we have previously used successfully in Veny et al. (2022) to perform a similar exercise where data near an island slope needed to be filled prior to volume transport calculations. This algorithm* accounts for the surrounding gradients to perform a realistic extrapolation. We will discuss this method in the revised version as well as include a mention of the bottom-triangle approach described by Thompson and Heywood (2008).

John D'Errico (2024). inpaint_nans (https://www.mathworks.com/matlabcentral/fileexchange/4551-inpaint_nans), MATLAB Central File Exchange.
*Following the suggestion by #Reviewer1: to account for the gaps near the sea bottom.*

We look forward to hearing back from you.

Sincerely,

Tania Pereira-Vázquez
* * *
POINT-BY-POINT RESPONSE

**REVIEWER #1**

**In this study seasonal variations of the Weddell Gyre's Western Boundary Current System are investigated. Due to the lack of continuous observational data in this harsh and remote environment, the authors primarily rely on three open-access reanalysis products (GLORYS12V1, GLORYS12V4 and GLBv0.08) for their transport analysis across the historical ADELIE transect and beyond (Expanded ADELIE transect). The reanalysis products are provided in different resolutions (horizontally and vertically), time spans and different forcing mechanisms incorporated. The authors main conclusions include seasonal variations in cross-transect transport of the WBCS significantly correlated to seasonal variations in wind stress.**

**I have several major concerns with this manuscript, but what really concerns me the most is the total negligence of the different bathymetry outputs that vary quite significantly for each of the reanalysis products. For example in Fig. 5 at 350 km off the continental shelf the difference in the vertical extent of the water column between GLORYS12V1 and GLBv0.08 is 1000 m. Specifically in the Southern Ocean the bathymetry products incorporated in the reanalysis products aren't reliable and do not represent troughs on the continental shelves, shelf break and slope sufficiently. Thus, the results of the cross-transect transports (top to bottom) calculations across multiple reanalysis products with different bathymetries can be impacted significantly, purely by different vertical extents. Thus, a direct comparison to observations is difficult without comprehensive discussion. This discussion and the caveats with respect to transport calculations that the different bathymetries bring have been left out entirely. Furthermore, it has been reported previously and also mentioned by the authors that some currents in the WBCS of the Weddell Gyre are bottom-intensified specifically at the continental slope, but bottom-triangles have not been included to improve transport calculations (as done by Thompson and Heywood (2008)).**

Response to the above paragraph: We agree about this point raised by the reviewer and are grateful he/she brings it to the table of discussion to improve our results. To this regard, we would like to note that a direct comparison with observations and reanalysis products is always challenging as the direct observations themselves count with limitations as well and in most studies they neither reach the bottom in all stations. This requires the above mentioned need to fill in the data gaps down to the sea floor, noting that not in all studies the same procedure to do this is applied. Bearing this in mind, we will make the reader aware about these limitations in the discussion and comparison of the estimates derived from the reanalysis products and the ones reported in the literature. Among the latter works, some modeling studies are included, which also adds uncertainty about the procedure followed to fill in the same gaps. For all these reasons, we understand that a clear and straightforward comparison will never be doable so we will address our discussion more carefully but confident that the results are worthy of being mentioned as they fall within the expected range of volume transport values. Furthermore, this is the approach traditionally followed by former studies (Matano et al., 2002; Neme et al., 2021, their Table 1) where estimates derived from modeling or reanalysis products are compared with existing observations, making the reader aware about the limitations we may be facing by doing so.

Lastly, we appreciate your observation regarding bottom-reaching transport. To address this, we have updated our volume transport estimates by applying the same technique as previously performed in Veny et al. (2022). Using the algorithm developed by D'Errico (2024), we retain original values and extrapolate over gaps accounting for the surrounding gradients, which is analogous to the bottom-triangles approach used by Thompson and Heywood (2008). We will indicate this approach, as well the bottom-triangles approach used by Thompson and Heywood (2008), in the revised version. Previously, the volume transport estimates for the total WBCS along the E-ADELIE transect in GLORYS2V4, GLORYS12V1, and GLBv0.08 were 34.93 Sv, 34.76 Sv, and 55.00 Sv, respectively. After filling in the gaps near the bottom, the updated estimates are 38.85 Sv, 37.43 Sv, and 61.24 Sv, respectively.

We agree that incorporating a bottom-reaching method improves the accuracy of volume transport calculations and makes the reanalysis products more comparable. Consequently, we have recalculated all transport values accordingly. Also, in the revised version we make the reader aware about the challenge that reanalysis products and models face when addressing bottom-intensified jets. As a preview, the reviewer is invited to have a look at the revised climatological vertical sections after filling in the gaps near the sea-bottom (see Fig. R1 by the end of this document).

Lastly, we recall here that although all new analyses have been performed also through the HYCOM-based product, from now on for brevity we will focus only on the NEMO-based product and observations, as explained in the preamble of this document of response. If the reviewers consider it necessary to show or indicate as well all the analyses regarding the HYCOM-based product, we will be happy to do so in a later document.

**Moreover, I would be hesitant to include GLBv0.08 at all (as there is no motivation given of why using it to begin with). To my understanding it does not provide good results or correlations (or at least it is not explained very well) and the time span and vertical resolution differ quite significantly from the other two products. Thus, I find a comparison of time-averaged parameters difficult if the time span differs between the different outputs. In that context, transports from time-averaged velocities of more than 20 years with observational data are not in the slightest comparable. By time-averaging, a lot of the vertical and horizontal structure is smoothed out and also seasonal and interannual variations (which are also quite strong in the Southern Ocean) are completely ignored. This comparison stands in contrast to your main result, where strongest transports occur during winter months. In my opinion, the authors have tried to cover to many questions at once without providing a detailed response to any of those questions. Thus, the 'storyline' is not clear to me.**

**Furthermore, I am missing a lot more detailed discussion on results with respect to differences between the reanalysis products, their caveats, sufficient reasoning for differences in results (other than resolution) and detailed comparison with previous studies. Specifically for differences in results a lot of the explanation provided is speculation and no evidence on the robustness of the results is provided. As you have nearly unlimited options when using reanalysis products, your study would really benefit from additional and detailed analysis and extensive discussion of caveats of the models and their variability in its entirety.**

Response to the above paragraphs:

We agree that the inclusion of GLBv0.08 broadens the possibilities of analysis in an unlimited scenario and the motivation behind our study was maybe not clear enough. In the revised version, we have addressed the new analyses proposed by the reviewers and find that the first revised version exceeded a suitable number of pages to deliver a clear message. For this reason, the revised version will not count with results from the HYCOM-based product.

Thus, in the revised version we do not only analyze the impact of the different spatial resolutions, which affect the multi-jet structure, but also the impact of the different vertical resolutions in capturing the definition of the bottom-intensified jets, and the impact of the sea-ice and wind forcings as one moves towards the ocean interior; it appears the wind stress forcing strongly modulate the strength and variability of the IWC (the major volume transport contributor of the Western Boundary Current System of the gyre) while closer to the coast the wind forcing appears less relevant and the thermohaline forcing prevails (sea-ice formation/melting).

We think that extending our analyses further beyond the above mentioned ones would require addressing modeling engineering aspects that are out the scope of this work. Thus, we hope that addressing the spatial and vertical resolution differences as well as the wind forcing role will set a more solid message, through which we conclude that the high resolution NEMO-based reanalysis product (GLORYS12V1) appears to respond the closest to the real ocean; understanding better the impact of horizontal and vertical grid resolutions as well as of the wind and sea-ice forcing. These findings support the use of the high resolution NEMO-based reanalysis product in future studies about the interannual variations of the WBCS of the Weddell Gyre. We think that it is important to report when global ocean reanalysis products in open access deliver ocean fields with some resemblance to the real ocean, also making aware to the scientific community about the limitations to bear in mind.

We hope the reviewer agrees that if we address now other analysis beyond those indicated above, the paper would certainly exceed a suitable number of pages to deliver a clear message. However, we are open to hear whether some specific additional test may be required.

Regarding the concern of the reviewer about the comparison of time-averaged estimates through different time periods between the reanalysis products, we understand the comment. In the revised version we treat this point more carefully and make the reader aware about the different time periods; however, climatological comparisons are a standard practice in oceanographic research to characterize seasonal patterns and not always the same time period is available. This is why time-averaging must involve several years of data (climatologies) in order to retain recurrent seasonal patterns. This is the case for our three reanalysis products, where we use at least 20 years of data. We agree an issue may exist if we would count with just a couple of years for one of the reanalysis products, since then interannual variability would play a strong role, but this is not the case.

Lastly, we would like to note it was not our intention to tell the reader that a straightforward comparison was doable between observations and climatologies from reanalysis products. By definition, the climatological view is always a smoothed scenario. However, we think it is still of value to report and discuss our climatologies against existing observations as long as they agree on the same order of magnitude. We do not claim anywhere in the text that the reanalysis product matches the in situ observations; differently, we indicate that estimates based on in situ observations approach the volume transport and ocean property ranges exhibited in the reanalysis products.

**Please find all major and minor comments in the attached pfd.** (See below what the reviewer attached).

**In general, I think this study is interesting. It does have potential for a future publication. However, there are many aspects with respect to methodology and caveats of the reanalysis products, which have not been addressed sufficiently. Plenty of open questions remain. In all fairness, I doubt that including an extensive additional analysis and discussion in the time frame given for major corrections would be possible. Therefore, I suggest to reject the manuscript at this stage, but encourage the authors to resubmit once major issues have been addressed in detail.**

We thank reviewer #1 for the careful reading of our work and for the time and effort dedicated to providing feedback about our research work. Now, we proceed to the point-by-point review of the submitted manuscript.

**Lines 9-35: The abstract can be significantly shortened. A lot of detail explaining the models can be removed here to keep a focus on the main results.**

We agree and will proceed to shorten the abstract in the final version according to the final comments by the reviewer #1 and #2.

**Line 34: How close? Is there a significant correlation between observations an model? How robust are these results?**

This statement is based on the following results:

- GLORYS2V4 does not reproduce the multijet structure of the WBCS because of its limited spatial resolution (the mesoscale at this latitude is not resolved). Also, the bottom intensified nature of the jets is not captured in spite of being the product (among the three of study) with the highest vertical resolution (see Fig. R1). This is again attributed to this reanalysis product not being capable to resolve the mesoscale nature of the multijet structure of the WBCS of the Weddell Sea.

- GLBv0.008 does reproduce the multijet structure of the WBCS of the Weddell Sea but does not reproduce the bottom intensified nature of the jets due to gappy data close to the sea floor of the continental slope (as already noticed by the reviewer). Furthermore, GLBv0.08 appears to overestimate the volume transport of the IWC as compared to estimates from ADCP observations reported in the literature, altimetry data (when computing surface volume transport), former modeling works and GLORYS data (this study). That overestimation exceeds 15 SV (even larger on occasions). For all the above reasons, and following the reviewer's suggestion, we have decided to drop the results from GLBv0.008 in the revised version of our manuscript.

- GLORYS12V1 does reproduce the multijet structure of the WBCS, the bottom intensified nature of the jets, and a range of volume transport variability which is within the order of magnitude of ADCP and former modeling-based estimates reported in the literature. To this regard, we will also make the reader aware about the fact that the comparison against model-based and observational-based estimates formerly reported in the literature is not straightforward as different model configurations, bathymetries and extrapolations methods come into play. However, we think the comparison we perform is still of value since even with all these circumstances, we find estimates from GLORYS12V1 are in agreement with former studies.

The revised version will contain the above notions and argued statements following the results.

**Lines 41: Remove roughly**

Agree, word removed.

**Lines 41: Could you show a map of the Weddell Gyre in its completeness?**

Thank you for your suggestion. We have revised Fig. 1 and now it includes an inset showing the full Weddell Sea basin. Regarding the sketch of main currents, we think it is important to zoom in over the western boundary as our study does not deal with the ocean dynamics of the entire gyre; differently, we focus on the Western Boundary Current System at the location where farther downstream, currents (and transported water masses) bifurcate and either abandon the basin or recirculate within the gyre.

**Line 46-48:  Rather long sentence. Suggest splitting in two.**
**Line 46: Fig. 1?**

We agree and have proceed to split the sentence in two, and refer to Fig. 1:

Before, these lines read:

Over this domain, a multi-jet structure is developed and water masses flowing within the western branch of the gyre either recirculate within the gyre or leave the basin towards the Bransfield Strait or the Scotia Sea, then flowing northward into the South Atlantic Ocean (Hellmer et al., 2005; Naveira Garabato et al., 2002).

Now, these lines read:

Over this domain, a multi-jet structure is developed (Fig. 1). Water masses flowing within the western branch of the gyre either recirculate within the gyre or leave the basin towards the Bransfield Strait or the Scotia Sea, then flow northward into the South Atlantic Ocean (Hellmer et al., 2005; Naveira Garabato et al., 2002).

**Line 53: Variability**

Agree, we have proceeded to change 'variations' to 'variability'.

**Line 54-55: I think it would be really nice if you could give a bit more reasoning for the chosen seasons. I acknowledge that Dotto has defined four seasons, but Zhang et al only two seasons (as widely known for the Antarctic region). The question is why is it relevant to define four seasons instead of two?**

We decided to define four seasons (three months each) based on the approaches by Dotto et al. (2021) and Zhang et al. (2011), who both provide a rationale for this seasonal division:

- Dotto et al. (2021) state: "The data sets were grouped in different seasons to allow a better representation of the region: summer (January to March), autumn (April to June), winter (July to September), and spring (October to December)."

- Zhang et al. (2011) note: "Following Kahl (1990) and Serreze et al. (1992), three-month seasons are defined as January–March (Arctic winter, Antarctic summer), July–September (Arctic summer and Antarctic winter), etc."

Not only because of the above works, but also based on our own results, we adopted the four seasons approach because the volume transport behavior in our study area shows four distinct patterns throughout the year, as depicted in Fig. 6 of the submitted manuscript.

To further demonstrate the impact of seasonal definitions, we have calculated volume transport means for the three products using both two-season and four-season definitions. The results indicate that defining only two seasons smooths out strongly the seasonal variations, to the point that accounting for their standard deviation to the winter and summer values, they approach closely, masking the natural mode of variability of the system regarding seasons. We think a finer seasonal definition based on 3 months provides a clearer and more accurate depiction of the volume transport variability within a climatological year.

- Volume transport using two seasons:

Summer (ONDJFM):
GLORYS2V4: 35.3 ± 3.8 Sv
GLORYS12V1: 34.9 ± 2.9 Sv

Winter (AMJJAS):
GLORYS2V4: 42.4 ± 1.3 Sv
GLORYS12V1: 39.8 ± 0.7 Sv

- Volume transport using four seasons

Summer (JFM):
GLORYS2V4: 33.5 ± 3.1 Sv
GLORYS12V1: 33.8 ± 3.0 Sv

Autumn (AMJ):
GLORYS2V4: 42.6 ± 1.6 Sv
GLORYS12V1: 40.0 ± 0.6 Sv

Winter (JAS):
GLORYS2V4: 42.2 ± 1.3 Sv
GLORYS12V1: 39.5 ± 0.8 Sv

Spring (SON):
GLORYS2V4: 37.1 ± 4.0 Sv
GLORYS12V1: 36.1 ± 2.8 Sv

Note that GLORYS2V4 and GLORYS12V1 are averaged from 1993 to 2020, corresponding to the temporal coverage of both products. Months defining each season are indicated by the capital letters between brackets as follows: J = January, F = February, M = March, A = April, etc.

**Line 68-69: Can you add specific number to the water mass characteristics. Some definitions vary so it would be good to have a proper definition of each water mass for your study to avoid misunderstandings.**

We appreciate the comment of the reviewer and understand the importance of clearly defining the water mass characteristics in our study region. We have included Tables R1 and R2 in the Appendix of this document of response to address this petition. This table will be added to the revised version of the submitted manuscript.

Lines 75-76:
An overview of the water masses which compose the multi-jet structure of the WBCS in the reanalysis products as compared to observations is provided in the Appendix, including their main characteristics (Fig. YY and Table YY). *Note: YY stands for the numbers of the corresponding figures and tables after adjustment in the revised version.*

Appendix:
To ensure clarity, we provide an overview of the characteristics of the main water masses encountered along the E-ADELIE transect. This summary is based on the literature and data from GLORYS12V1; these ocean property ranges are in agreement with those found for GLORYS2V4.

*See Table R1 and Table R2.*

**Line 69-70: This is not quite right. Strong winter cooling and brine release during sea ice formation produces dense shelf water. Antarctic Bottom Water is formed when the dense shelf water spills down the continental slope and mixes with circumpolar deep water.**

Agree, we have proceeded to better explain the process occurring in this area, attending this point raised also by R#2.

Before, these lines read:

Across this front there is a rapid change in temperature and salinity due to the interaction between the colder and fresher waters of the Antarctic continental shelf, formed as a result of sea ice formation and melting processes, and the relatively warm and saline waters sourced by the Antarctic Circumpolar Current (ACC) (Thompson & Heywood, 2008; Vernet et al., 2019). The former water mass is known as Antarctic Surface Water (AASW), while the latter water mass is a modified Circumpolar Deep Water known as Warm Deep Water (WDW). Down the continental slope, dense Antarctic Bottom Water (AABW) is formed following brine rejection in the upper ocean surface.

Now, these lines read:

Across this front there is a rapid change in temperature and salinity due to the interaction between the colder and fresher waters of the Antarctic continental shelf, formed as a result of sea ice formation and melting processes, and the relatively warm and saline waters sourced by the Antarctic Circumpolar Current (ACC) entering the gyre from the eastern part (Thompson & Heywood, 2008; Vernet et al., 2019). The former water mass is known as Antarctic Surface Water (AASW), while the latter water mass is a modified Circumpolar Deep Water known as Warm Deep Water (WDW) within the Weddell Sea (Schröder & Fahrbach, 1999). Over the continental shelf, dense shelf water produced during sea ice formation cascades down the continental slope and mixes with WDW. This process allows the dense water to reach the ocean bottom and continue its pathway through the global ocean. This dense bottom water is known as Antarctic Bottom Water (AABW) (Muench & Gordon, 1995; Stewart & Thompson, 2012).

**Line 71:  I agree with you. However, the ASF is not entirely circumpolar. It is missing along the West Antarctic Peninsula and large parts of the Bellingshausen Sea. Rather than generalizing you can be more specific in which regions this is actually relevant.**
**In the WAP and Bellingshausen Sea it is the southern boundary of the ACC that is closest to the shelf break.**

Agree, we specify the areas where ASF is absent.

Before, these lines read:

Thus, the ASF plays a crucial role in the exchange of heat, salt, and nutrients between the deep ocean and the Antarctic continental shelf waters with important implications for the distribution of marine ecosystems and sea-ice dynamics, where the WDW also conditions the melting of ice shelves.

Now, these lines read:

Thus, the ASF plays a crucial role in the exchange of heat, salt, and nutrients between the deep ocean and the Antarctic continental shelf waters, with important implications for the distribution of marine ecosystems and sea-ice dynamics (Vernet et al, 2018). In these regions, the WDW also influences the melting of ice shelves. However, the ASF is not entirely circumpolar; it is interrupted by the Antarctic Peninsula, separating the Pacific and Atlantic sectors of the Southern Ocean (Thompson et al., 2018). Specifically, it is absent along the West Antarctic Peninsula and large parts of the Bellingshausen Sea.

**Line 73-75:  does it have a name?**

These lines read:

Lastly, towards the ocean interior, another jet is found associated with the WF, which holds a major frontal system linked to high mesoscale variability in the form of eddies and meanders (Heywood et al., 2004; Thompson & Heywood, 2008).

We find that in the literature the discussion is always held in terms of the jet associated with the Weddell Front and on several occasions the authors refer to the dynamic feature with the same name as the hydrographic feature. We know this may be confusing, one feature is the hydrographic front, and another feature is the jet associated to the front but, so far, they are used indifferently in the literature. We may introduce the name Weddell Jet to make a distinction but, on the other hand, we think the use of WF is already accepted in the literature to refer to the jet and do not find a strong reason to make changes about it.

**Line 77:  The figure really lacks in resolution. Please align the ticks of the colorbar with color transitions (e.g. by adjusting the number of colors in your color map). This goes for all following colorbars in other Figures.**
**It is not neccessary for the global map to be this big. Its covering half the ACC current in Fig. 1. I suggest doing a stereographic plot from the Southern Ocean region and mark the area of interest with a box. Without adding axis labels. This should give a nice and simple overview of where the region of interest is located on a 'Southern Ocean Perspective'.**

We agree with the comments, and have proceeded with the suggested changes to Fig. 1 and 2 of the submitted manuscript. The revised figures are shown at the end of this document as Fig. R2 and R3, respectively.

**Line 77:  Which bathymetry is used here?**

We use the *satbath* function from Matlab to extract bathymetry: *satbath* reads the global topography file for the entire world from topo_8.2.img, which is a geospatial data file that contains topographic and/or bathymetric information in a raster image format. *topo_8.2.img* originates from Smith and Sandwell (1997).

In the revised manuscript we will add this reference to the caption of Figure 2 as follows:

Figure 2. Bathymetric map of the north-western sector of the Weddell Gyre depicting the study area with indication of the ADELIE transect … The bathymetry originates from topo_8.2.img (Smith & Sandwell, 1997).

Reference:

Smith, W. H., & Sandwell, D. T. (1997). Global sea floor topography from satellite altimetry and ship depth soundings. *Science*, *277*(5334), 1956-1962.

**Line 81: This would need to go in the text not in the caption.**

Agree, sentence removed from the caption.

**Line 82: it just makes this sentence really complicated.**

Agree, sentence removed from the main text.

**Line 86: Are these authors the only ones who have considered this?**

These authors are not the only ones studying how resolution affects the Weddell Gyre in ocean models, but is the latest work addressing this topic. Other articles addressing this issue are:

Renner, A. H. H., Heywood, K. J., and Thorpe, S. E.: Validation of three global ocean models in the Weddell Sea, Ocean Modelling, 30, 1–15. https://doi.org/10.1016/j.ocemod.2009.05.007, 2009.

Neme, J., England, M. H., and Hogg, A. M.: Seasonal and interannual variability of the Weddell Gyre from a high-resolution global ocean-sea ice simulation during 1958–2018, J. Geophys. Res: Oceans, 126, e2021JC017662. https://doi.org/10.1029/2021JC017662, 2021.

Both articles and also the article of Styles et al 2023 agree that increasing resolution of the models to be eddy-permitting is a good choice. We have added this notion in the revised version as follows:

Before, these lines read:

When considering a basin situated at high latitudes, it is not unexpected to anticipate that the horizontal resolution of the model will most likely play a crucial role in the realistic simulation of the ocean dynamics of the Weddell Sea. Recently, an idealized model of the Weddell Sea gyre was investigated to test how inter-model variability can originate from differences in the horizontal resolution of the ocean model (Styles et al., 2023).

Now, these lines read:

When considering a basin situated at high latitudes, it is not unexpected to anticipate that the horizontal resolution of the model will most likely play a crucial role in the realistic simulation of the ocean dynamics of the Weddell Sea (Neme et al., 2021; Renner et al., 2009). Recently, an idealized model of the Weddell Sea gyre was investigated to test how inter-model variability can originate from differences in the horizontal resolution of the ocean model (Styles et al., 2023).

**Line 88: As they have only considered an idealized model. How applicable are these results to your study?**

We believe it is important to provide readers with a comprehensive overview of the various studies conducted in the study area, which employ different methods and yield diverse results. This contextualization allows for a fair comparison and an insightful discussion of the results in relation to the existing body of knowledge. Therefore, our aim is not to make a direct comparison but to present a state-of-the-art introduction of the different approaches used to study this area.

**Line 91: Use not instead of far from, and negligible instead of neglectable**

Agree, we changed both words in the main text.

**Line 100: I am missing parts of motivation here. Why is this relevant to your study?**

As we mentioned in response to line 88 comment, our aim is to present a state-of-the-art introduction of the different approaches used so far to study this area.

**Line 105: remove actually**

We agree, word removed.

**Line 107: Again, why is this relevant to your study. I am ok with you referring to section 3 for details. But mentioning why your study is important to consider the seasonality hast to be highlighted more to strengthen your research.**

Our study addresses the challenge of obtaining a year-round characterization of the NW Weddell Gyre. We present for the first time a climatology of the WBCS of the Weddell Gyre based on open-access reanalysis products that assimilate in-situ data, permitting us to describe the natural mode of oscillation of the system.

In these lines we are just making the point that the work we cite also considers important to account for the strong seasonal variability present in the study region.

We make our point stronger in the following lines, which will be included in the revised manuscript by the end of the Introduction section.

Before, these lines (147-152) read:

Ultimately, we designed this work to characterize the seasonal variability, in location and transport, of the multi-jet structure governing the western Weddell Sea from a climatological point of view based on two open-access reanalysis products, supporting our analyses with direct velocity measurements, altimetry data and a complete review of the state-of-the-art knowledge in the literature. Furthermore, we also target to evaluate the goodness and convergence of these two open-access reanalysis products with different resolutions in delivering similar and coherent circulation patterns, both qualitatively and quantitatively.

Now, these lines read:

Ultimately, we aim to characterize the seasonal variability of the multi-jet structure governing the Western Boundary Current System of the Weddell Gyre in terms of spatial distribution and volume transport dynamics. We do this from a climatological point of view based on two open-access reanalysis products, supporting our analyses with direct velocity measurements, altimetry data and a complete review of the state-of-the-art knowledge in the literature. The motivation to characterize the seasonal variability is to provide a climatological background for future studies about the year-to-year variability in this region, the baseline for any solid analysis about variations at longer time-scales. Furthermore, we assess how the spatial and vertical resolution of the two reanalysis products (GLORYS2V4 and GLORYS12V1) affect the representation of the current system of study.

**Line 109: which scenario?**

We have rephrased these lines for clarity.

Before, these lines read:

This scenario fuels the motivation behind this work. We do not use an idealized model, neither a climate model but two global ocean circulation reanalysis products with different horizontal resolutions and a set of in situ and remotely-sensed observations.

Now, they read:

The scarcity of year-round estimates of the Weddell Gyre transport based on observations fuels the motivation behind this work. We do not use an idealized model, neither a climate model but two global ocean circulation reanalysis products with different horizontal resolutions and a set of *in situ* and remotely-sensed observations.

**Line 109-111: as you don not use an idealized model. How comparable are these results to your study? Clearly bathymetry is important for transport calculations as well?**

Idealized models are always the first step before trying to address a more complex system. As previously explained, we do not bring the idealized model to the discussion for comparison with our results but to provide background about former studies and what we have learned from them so that we can better discuss what we observe. Idealized models provide insights about the underlying major roles governing differences observed in the real ocean. In several former responses we have tried to rephrase our text and leave this point clear; however, if the reviewer considers this is still not clear and has a suggestion about how we should rephrase these notions, we will be happy to do so.

We address the issue of the bathymetry and its importance for volume transport later (see response to line 258). Regarding the bathymetry source used by each product, information is shown in Table 1 of the submitted manuscript.

**Line 117: In which context?**

We refer to the same scenario explained before, in the answer of line 109. Because we understand it may be confusing, we have removed the starting phrase. We think the former paragraph already provides enough context.

Before, these lines read:

In this context, the historical transect known as ADELIE (Antarctic Drifter Experiment: Links to Isobaths and Ecosystems), depicted in Figs. 1 & 2, constitutes the most convenient reference frame to assess the ocean dynamics in the northwest of the Weddell Gyre (Thompson & Heywood, 2008).

Now, they read:

The historical transect known as ADELIE (Antarctic Drifter Experiment: Links to Isobaths and Ecosystems), depicted in Figs. 1 & 2, constitutes the most convenient reference frame to assess the ocean dynamics in the northwest of the Weddell Gyre (Thompson & Heywood, 2008).

**Line 123: Replace 'us from counting with year-round observations.' by 'continuous observations throughout the year.'**

Agree, sentence replaced.

**Line 124-125: Do you mean: remotely-sensed observations representing the surface ocen dynamics are strongly ... ??**

We just meant that during the sea-ice season, remotely-sensed observations are a major challenge.

Before, these lines read:

Also, remotely-sensed observations about the surface ocean dynamics are strongly limited during the sea-ice seasons because the presence of the sea-ice cover poses a major challenge for satellite sampling.

Now, these lines read:

Also, remotely-sensed observations about the surface ocean dynamics are limited during the sea-ice seasons because the presence of the sea-ice cover poses a major challenge for satellite sampling.

**Line 125: Do you have reference for this statement?**

Yes, in the revised version we are providing the following reference for this line:

Kwok, R., & Cunningham, G. F. (2015). Variability of Arctic sea ice thickness and volume from CryoSat-2. *Philosophical Transactions of the Royal Society A: Mathematical, Physical and Engineering Sciences*, 373(2045), 20140157. https://doi.org/10.1098/rsta.2014.0157.

**Line 125: try to avoid this type of phrasing as much as possible as it is 'spoken language'**

Before, these lines read:

Hence, the use of reanalysis products becomes crucial so we can explore the governing year-round ocean dynamics in this region without time gaps.

Now, these lines read:

Hence, the use of reanalysis products becomes crucial so that the governing year-round ocean dynamics in this region can be explored without time gaps.

**Line 129: so the extended Adelie transect is only used for the reanalysis products?**

We appreciate this point and would like to clarify the use of the E-ADELIE transect in our study. The E-ADELIE transect is established starting from the coordinates of the first station in the original ADELIE transect. While a similar but larger section is found in the literature, under the name of 'SR04 WOCE' transect (see figure below), our specific transect location differs slightly. However, we agree that it is important to reference the SR04 WOCE section, as noted by R#2. Therefore, we will revise the main text to clarify that the E-ADELIE transect is a portion of the existing SR04 WOCE section.This adjustment will provide better context for our readers while preserving the terminology used throughout the manuscript.

[Figure]

Figure by Rodrigo Kerr et al., 2005 - Location of WOCE SR04 repeat hydrographic section.

Source: Kerr, Rodrigo & Mata, Mauricio & Garcia, Carlos. (2005). Optimum Multiparameter Analysis of the Weddell Sea Water Mass Structure. Clivar Exchanges. 10. 33-35.

Before, lines 238-246 of the submitted manuscript read:

To characterize the variability of the WBCS of the Weddell Gyre we focus on a key location, the historical ADELIE transect (Fig. 2), where the current system is well defined and has been previously described as a multi-jet structure current system (Thompson & Heywood, 2008). The ADELIE transect is located northeast of the AP, where the Weddell waters may either turn around the Antarctic Peninsula towards the Bransfield Strait, leave the gyre circulation towards the Scotia Sea and the South Atlantic Ocean, or recirculate within the Weddell Gyre. Different from the traditional ADELIE transect, we analyse a version of that one which extends farther oceanward into the gyre interior and which we name E-ADELIE transect. This transect aims to account for the dynamics of the western branch of the Weddell Gyre to its full extent. We address the comparison of the horizontal and vertical structure of the WBCS jets as depicted by the two reanalysis products, SADCP measurements and altimetry data.

Now, these lines read:

To characterize the variability of the WBCS of the Weddell Gyre, we focus on a key location, the western part of the historical SR04 WOCE section. The western portion of this section was previously studied by Thompson & Heywood (2008), who refer to this section as the ADELIE transect (Fig. 2), where the current system is well defined and described as a multi-jet structure current system. Different from the ADELIE transect, we analyze a version that extends farther oceanward into the gyre interior, which we name the E-ADELIE transect. Here, E-ADELIE refers to the western portion of the SR04 WOCE section. This transect aims to capture the dynamics of the western branch of the Weddell Gyre in its entirety, including the IWC jet. We address the comparison of the horizontal and vertical structure of the WBCS jets as depicted by the two reanalysis products, SADCP measurements, and altimetry data.

**Line 133:  again which bathymetry has been used here?**

We use the *satbath* function from Matlab to extract bathymetry: *satbath* reads the global topography file for the entire world from topo_8.2.img, which is a geospatial data file that contains topographic and/or bathymetric information in a raster image format. *topo_8.2.img* originates from Smith and Sandwell (1997).

In the revised manuscript we will add this reference to the caption of Figure 2 as follows:

Figure 2. Bathymetric map of the north-western sector of the Weddell Gyre depicting the study area with indication of the ADELIE transect … The bathymetry originates from topo_8.2.img (Smith & Sandwell, 1997).

Reference:

Smith, W. H., & Sandwell, D. T. (1997). Global sea floor topography from satellite altimetry and ship depth soundings. *Science*, *277*(5334), 1956-1962.

**Line 157: I suggest revising the number of sub- und subsubsections. I see the point of this providing structure to the paper, but it also takes up a lot of space. I think it is much easier to read without interruptions when there is less subsections as you can easily differentiate the different data sets by starting a new paragraph. I realize this might be personal preference, so there are no obligations from my side, but its worth a thought. Suggesting the following:**

**2 Data and methodology**
**2.1 Data**
**2.2 Methodology**

We agree, in the revised version we have reduced the number of subsections and use now the suggested structure.

**Line 164-166: This is just repetition at this point. I suggest removing both the preambles entirely. They just make it unnecessarily lengthy to read and are not really needed to understand what is explained. You can merge the preambles with the first pargraph of the subsections where you actually start explaing the data etc. This way you avoid repetition.**

Agree, we have removed the paragraph to avoid repetition.

**Line 170: Well then table 1 should follow immediately after this sentence. Not that the readers who might not be familiar with these reanalysis products don't now anything about the model details at this point.**

We agree and have changed the position of Table 1.

**Line 172: This type of sentence beginning does not make any sense in this context. I suggest to remove it.**

We agree and have removed this beginning of the sentence.

**Line 172-173: this belongs to the data availability statement not here. same here**

We agree and have moved this information to the *Data Availability Section,* required by the Journal.

All data we used in this investigation are publicly accessible. Global NEMO-based reanalysis products GLORYS2V4 and GLORYS12V1 are available at https://doi.org/10.48670/moi-00024 (last access 29/05/2024) and https://doi.org/10.48670/moi-00021 (last access 08/02/2024), respectively. Data of Sea Ice Coverage from GLORYS products are also accessible at the same link as the products (last access 13/06/2024). Regarding wind products, those forcing GLORYS products are ERA-interim and ERA5 datasets, which are available in open access at https://www.ecmwf.int/en/forecasts/dataset/ecmwf-reanalysis-interim (last access 29/05/2024) and https://cds.climate.copernicus.eu/cdsapp#!/dataset/reanalysis-era5-single-levels?tab=overview (last access 29/05/2024).
Regarding altimetry data, they are available at https://doi.org/10.48670/moi-00148 (last access 19/12/2022).

Both SADCP data are free accessible, LG0003a of R/V Laurence M. Gould cruise data are available at http://adcp.ucsd.edu/lmgould/lmgadcp/year2000.html#lg0003a (last access 14/02/2024), and NBP0106 cruise of R/V Nathaniel B. Palmer cruise data are available at https://currents.soest.hawaii.edu/nbpalmer/nbp0106/nb150/webpy/ (last access 29/02/2024).
Finally, in situ data from Cruise SOS-Climate II of RV Ary Rongel (Fig. A1 from appendix), are accessible at https://doi.pangaea.de/10.1594/PANGAEA.864578 (last access 26/05/2021).

**Line 173: Both? How about: Both reanalysis products are global ....**
**Line 174: Remove respectively. provided by instead of developed**

We appreciate the suggestions and so we have reformulated the sentence.

Before, these lines read:

They are global eddy-permitting and eddy-resolving ocean products of reanalysis, respectively, developed by the Copernicus Marine Environment Monitoring Service (CMEMS).

Now, these lines read:

Both ocean reanalysis products are global eddy-permitting and eddy-resolving respectively, provided by the Copernicus Marine Environment Monitoring Service (CMEMS).

**Line 175: This also doesn't make any sense here. Remove.**

We agree and have removed 'thus'.

**Line 175: in your region of interest how many km is that respectively? Give the reader some numbers that are easily accessible. Remove respectively**

Agree, we have added this information to the referred line.

GLORYS2V4 and GLORYS12V1 consist of daily data with 0.25° and 0.08° of horizontal resolution, corresponding to ~15.9 km and ~5.4 km over the study region, approximately.

**Line 176-178: Spoken language. Remove 'Bearing this in mind'. This sentence is very confusing. Can you rephrase?**

We agree and have reformulated these lines.

Before, these lines read:

Bearing this in mind, at the regional scale of our high latitude study area, we must note that GLORYS2V4 data do not solve the mesoscale while GLORYS12V1 data does resolve the mesoscale as an eddy-solving ocean product.

Now, these lines read:

At the regional scale of our high latitude study area, it is important to note that GLORYS2V4 does not capture mesoscale features. In contrast, GLORYS12V1 is an eddy-resolving ocean product and does capture mesoscale dynamics.

**Line 178-179: Both ocean components are. This quite an outdated product. Did you compare atmospheric forcing between newer products such as ERA5 ? Also data availability statement. You must also include the date of last access.**

Agree, we have reformulated these lines and moved thse information to the availability statement.

Before, these lines read:

They are both based on the NEMO platform (Nucleus for European Modelling of the Ocean) and are atmospherically forced with ERA-Interim (https://www.ecmwf.int/en/forecasts/dataset/ecmwf-reanalysis-interim).

Now, these lines read:

Both ocean products are based on the NEMO platform (Nucleus for European Modelling of the Ocean) and are atmospherically forced with ERA-Interim until its discontinuation in 2019, after which ERA5 forcing fields have been applied starting from January 2019.

**Line 181: I don't hink you need to start a new paragraph here. This information still belongs to the previous paragraph. as you started a new paragraph it is better to keep using GLORYS12V1 etc as before. It provides consistency for the reader. same here**

We appreciate the suggestion and, for clarity, we have placed the referred sentences in the same paragraph.

**Line 184: potential temperature and practical salinity. Put units as well. Just using the terms temperature and salinity is not enough if they are not defined properly.**

Agree, we have added the information required on the main text.

potential temperature (ºC), sea water salinity ($10^{-3}$)

**Line 185: well which ones? are these data used?**

In the revised version we use Sea Ice Area Fraction according to the request of the reviewer #2. The revised lines read:

These databases deliver several ocean properties such as thermodynamic variables (temperature, salinity), dynamic variables (meridional and zonal total velocities, u and v, derived from sea surface height) as well as Sea Ice Area Fraction.

**Line 186: I am getting a bit confused here. It is the first time that this simulation is mentioned. Can you explain why you are using a third one? I am missing a valuable reasoning for using this simulation in comparison to the other ones!**

We have removed the paragraph between lines 186-192 since in the revised version we are not including results based on the HYCOM-based product.

**Line 188: These (plural)**

This text has been removed as indicated in the previous response.

**Line 210: where is the wind forcing?**

We specified the wind forcing details in Section 2.2.3 of the submitted manuscript. However, following your comment, we have also added the wind forcing information to Table 1 for further clarity.

**Line 224: What are you using these data for?**

We use SADCP data to support the horizontal spatial structure of the jets composing the Western Boundary Current System of the Weddell Gyre as captured from altimetry and reanalysis. Also, we use the SADCP to confirm the main features of the previously unreported IWC. We make these points clearer in the revised version by adding this information.

Before, these lines read:

We use data from two oceanographic cruises, pertaining to two different years and seasons.

Now, these lines read:

We use direct velocity measurements from two oceanographic cruises, collected from two different years and seasons, to support the spatial distribution of the jets as captured in altimetry and reanalysis data. Also, the in situ measurements are used to support further the first characterization of the IWC.

**Line 240-243: This has been mentioned before. Repetition.**

Agree, we have removed the sentence.

**Line 244-245: This extention of the historical ADELIE transect aims to ...**

We reformulate the 2.2.3 section for clarity, attending also to request from R#2.

Before, these lines (237-246) read:

2.2.1 The E-ADELIE transect

To characterize the variability of the WBCS of the Weddell Gyre we focus on a key location, the historical ADELIE transect (Fig. 2), where the current system is well defined and has been previously described as a multi-jet structure current system (Thompson & Heywood, 2008). The ADELIE transect is located northeast of the AP, where the Weddell waters may either turn around the Antarctic Peninsula towards the Bransfield Strait, leave the gyre circulation towards the Scotia Sea and the South Atlantic Ocean, or recirculate within the Weddell Gyre. Different from the traditional ADELIE transect, we analyse a version of that one which extends farther oceanward into the gyre interior and which we name E-ADELIE transect. This transect aims to account for the dynamics of the western branch of the Weddell Gyre to its full extent. We address the comparison of the horizontal and vertical structure of the WBCS jets as depicted by the two reanalysis products, SADCP measurements and altimetry data.

Now, these lines read:

2.2 Methods

To characterize the variability of the WBCS of the Weddell Gyre, we focus on a key location, the western part of the historical SR04 WOCE section. The western portion of this section was previously studied by Thompson & Heywood (2008), who refer to this section as the ADELIE transect (Fig. 2), where the current system is well defined and described as a multi-jet structure current system. Different from the ADELIE transect, we analyze a version that extends farther oceanward into the gyre interior, which we name the E-ADELIE transect. Here, E-ADELIE refers to the western portion of the SR04 WOCE section. This transect aims to capture the dynamics of the western branch of the Weddell Gyre in its entirety, including the IWC jet. We address the comparison of the horizontal and vertical structure of the WBCS jets as depicted by the two reanalysis products (GLORYS2V4 and GLORYS12V1), SADCP measurements, and altimetry data.

**Line 245:  but you used three though?**

In the revised version we are removing results from the HYCOM-based product. Accordingly, the new lines read:

We address the comparison of the horizontal and vertical structure of the WBCS jets as depicted by the two reanalysis products (GLORYS2V4 and GLORYS12V1), SADCP measurements, and altimetry data.

**Line 248: Replace namely to 'defined as'**

Agree, replaced.

**Line 253: The rotation of vectors should be assumed as basic knowledge and does not need to be included here. Your description of how and why your transect velocities are rotated is sufficient.**

We agree. We have removed the requested text.

**Line 258: Thompson and Heywood (2008) specifically make use of bottim triangle calculation to estimate the bottom outflow, which was found quite significant for transport calculations. Have you incorporated bottom triangles in your calculations? What difference does it make.**

We have filled in the gaps near the bottom-sea accounting for the bottom-intensified jets. To do so we have used an algorithm available in MATLAB code that we have previously used successfully in Veny et al. (2022) to perform a similar exercise where data near an island slope needed to be filled prior to volume transport calculations. This algorithm* accounts for the surrounding gradients to perform a realistic extrapolation. We will discuss this method in the revised version as well as include a mention of the bottom-triangle approach described by Thompson and Heywood (2008).

John D'Errico (2024). inpaint_nans
(https://www.mathworks.com/matlabcentral/fileexchange/4551-inpaint_nans),
MATLAB Central File Exchange.

We applied this function in segments, taking pairs of stations to ensure the function's good performance and accurate results. To illustrate the impact, we compared below the mean volume transport calculated between 1993-2020 for GLORYS2V14 and GLORYS12V1, before and after applying the bottom extrapolation (corresponding to the vertical section in Fig. 5 of the submitted manuscript and Fig. R1 of this document):

Before bottom extrapolation:
GLORYS2V4: *34.93 Sv*
GLORYS12V1: *34.77 Sv*

After bottom extrapolation:
GLORYS2V4: *38.85 Sv*
GLORYS12V1: *37.43 Sv*

Given the observed differences, and following the reviewer's request, we applied this method to all our calculations. Therefore, all the results presented in this document of response and in the revised document are bottom-reaching, ensuring a more accurate representation of the volume transport. We will also add the explanation of this extrapolation method in section 2.2.2 Volume transport.

Please, see the revised version of Fig. 5 of the submitted manuscript (referred in this document of response as Fig. R1 by the end of the document), following suggestions of R#1.

**Line 276: both equations 4 and 5 are unnecessary in my eyes- This can be sufficiently explained in a couple of sentences.**

In the revised version we do not rotate the wind components but rather average spatially the wind field covering the full basin, as suggested by the reviewer #1. For this reason, equations 4 and 5 are not anymore in the revised manuscript.

**Line 286: Do you have a figure to show this? I don't quite understand this reasoning.**

In response to your comment, we revisited our wind calculations and adjusted our approach to align with the statement of 'basin-scale calculations' (line 278). We expanded the area of integration for wind stress and included wind stress curl calculations as suggested by the reviewer #1 in comment to line 540. To clarify this, we have proceeded to add the computational details and a figure illustrating the area for spatial averaging to the appendix (Fig. R4), in addition to modifying the main text and adding wind stress curl calculations in the wind stress section (2.2.3). We have also proceeded to add the seasonal wind stress curl in Fig. 6 of the submitted manuscript (now Fig. R5) and its corresponding correlations with volume transport in Table 3 of the submitted manuscript (now Table R3). Lastly, we also took into account the sea ice area fraction to calculate wind stress, as suggested R#2.

Before, these lines (264-291) read:

We calculate wind stress as follows, using the formula proposed by Kara et al. (2013):

$$\boldsymbol{\tau} = \boldsymbol{\rho} \cdot \mathbf{U}_{10}^2 \cdot \mathbf{C_D} \quad (3)$$

where ρ represents the air density (1.2 kg m-3); $U10 = \sqrt{u10'2 + v10'2_{f0}}$ is the wind speed at 10 m above the surface (with v10' and v10' denoting rotated eastward and northward velocity components, respectively); and, CD is the drag coefficient, which is a function of wind speed, U10.
To ensure alignment with volume transport calculations, we compute cross- and along-ADELIE wind velocities. v10' and u10' are utilized to calculate U10, maintaining the same rotation angle (α) through the expressions:

$$v_{10}' = v_{10} * cos(\boldsymbol{\alpha}) - u10 * sin(\boldsymbol{\alpha}) \quad (4)$$
$$u_{10}' = v_{10} * sin(\alpha) + u10 * cos(\alpha) \quad (5)$$

We conduct basin-scale calculations to assess the average wind patterns across our study area. Subsequently, we analyse the relationship between the seasonal cycles of wind and volume transport, considering wind as a potential influencing factor. To do this, we delineate the boundaries of the study are as follows: the northern boundary corresponds to a distinct change in wind direction at 64ºS; moving southward, the limit is established by identifying the point where the signal of the WBCS becomes evident, which occurs at 74ºS; the western boundary is demarcated by the Antarctic Peninsula (AP) at 62ºW, while the eastern boundary is determined by an observable shift in both wind and current directions at 38ºW. The products employed for wind stress computation derived from the two reanalysis products used in this study. GLORYS2V4 and GLORYS12V1 use ERA-interim and ERA5 datasets, respectively. Since ERA-interim was discontinued in 2019, ERA5 forcing fields have been applied starting from January 2019, accessible at https://www.ecmwf.int/en/forecasts/dataset/ecmwf-reanalysis-interim and https://cds.climate.copernicus.eu/cdsapp#!/dataset/reanalysis-era5-single-levels?tab=overview. GLBv0.08 uses NCEP-CFS and NCEP-CFSV2 products. NCEP-CFS was discontinued in 2010. The corresponding data sets are accessible at https://www.hycom.org/dataserver/ncep-cfsr and https://www.hycom.org/dataserver/ncep-cfsv2.

Now, these lines read:

We calculate wind stress as follows, using the formula proposed by Lüpkes et al. (2005):

$$\tau = \rho \cdot \mathbf{U}_{10}^2 \cdot Ci \quad (2)$$

where ρ represents the air density (1.2 kg m$^{-3}$); U10=$\sqrt{(\mathbf{U'}_{10}^2 + \mathbf{V'}_{10}^2)}$ is the wind speed at 10 m above the surface (with u'$_{10}$ and v'$_{10}$ denoting rotated eastward and northward velocity components, respectively); and, Ci is the drag coefficient calculated following Lüpkes et al. (2005), after specifying the sea ice concentration.

Wind stress curl is calculated after obtaining the $\tau_x$ and $\tau_y$ components of the wind stress, accounting for sea ice over the Weddell Basin. Basin-scale calculations are performed to assess the average wind patterns across our study area (see Fig. YY in the appendix). We then analyze the relationship between the seasonal cycles of wind and volume transport, considering wind as a potential influencing factor.

**Line 285-291: this belongs to the data set description. this again belongs in the data availabilty statement. are those wind products? also data descritption though. again data availability statement.**

We agree with the comment by reviewer #1, and include this information in the data description section, and also move the corresponding information to the data availability section.

Before, these lines read:

The products employed for wind stress computation derived from the two reanalysis products used in this study. GLORYS2V4 and GLORYS12V1 use ERA-interim and ERA5 datasets, respectively. Since ERA-interim was discontinued in 2019, ERA5 forcing fields have been applied starting from January 2019, accessible at https://www.ecmwf.int/en/forecasts/dataset/ecmwf-reanalysis-interim and https://cds.climate.copernicus.eu/cdsapp#!/dataset/reanalysis-era5-single-levels?tab=overview. GLBv0.08 uses NCEP-CFS and NCEP-CFSV2 products. NCEP-CFS was discontinued in 2010. The corresponding data sets are accessible at https://www.hycom.org/dataserver/ncep-cfsr and https://www.hycom.org/dataserver/ncep-cfsv2.

Now, these lines read:

The products employed for wind stress computation derived from the two reanalysis products used in this study. GLORYS2V4 and GLORYS12V1 use ERA-interim and ERA5 datasets. Since ERA-interim was discontinued in 2019, ERA5 forcing fields have been applied starting from January 2019.

The *Data Availability Section,* required by the Journal, reads:

All data we used in this investigation are publicly accessible. Global NEMO-based reanalysis products GLORYS2V4 and GLORYS12V1 are available at https://doi.org/10.48670/moi-00024 (last access 29/05/2024) and https://doi.org/10.48670/moi-00021 (last access 08/02/2024), respectively. Data of Sea Ice Coverage from GLORYS products are also accessible at the same link as the products (last access 13/06/2024). Regarding wind products, those forcing GLORYS products are ERA-interim and ERA5 datasets, which are available in open access at https://www.ecmwf.int/en/forecasts/dataset/ecmwf-reanalysis-interim (last access 29/05/2024) and https://cds.climate.copernicus.eu/cdsapp#!/dataset/reanalysis-era5-single-levels?tab=overview (last access 29/05/2024).
Regarding altimetry data, they are available at https://doi.org/10.48670/moi-00148 (last access 19/12/2022).
Both SADCP data are free accessible, LG0003a of R/V Laurence M. Gould cruise data are available at http://adcp.ucsd.edu/lmgould/lmgadcp/year2000.html#lg0003a (last access 14/02/2024), and NBP0106 cruise of R/V Nathaniel B. Palmer cruise data are available at https://currents.soest.hawaii.edu/nbpalmer/nbp0106/nb150/webpy/ (last access 29/02/2024).
Finally, in situ data from Cruise SOS-Climate II of RV Ary Rongel (Fig. A1 from appendix), are accessible at https://doi.pangaea.de/10.1594/PANGAEA.864578 (last access 26/05/2021).

**Line 294: This is spoken language. Rephrase. also I don't understand what you are referring to with 'this latter work'**

'In this latter work' referred to the work by Gordon et al 1981. However, we have proceeded to reformulate the paragraph.

Before, these lines read:

The applicability of Sverdrup dynamics for calculating the Weddell Gyre's transport is not straightforward and has been questioned for decades (Gordon et al. ,1981). In this latter work though, the authors estimated the Weddell Gyre transport in ~76 Sv, using wind stress data and applying Sverdrup balance.

Now, these lines read:

The applicability of Sverdrup dynamics for calculating the Weddell Gyre's transport is not straightforward and has been questioned for decades, as in Gordon et al. (1981), where the Weddell Gyre transport was estimated in ~76 Sv using wind stress data and applying Sverdrup balance.

**Line 295: well 1991 und 1998 is not very recent is it?**

We meant more recent than Gordon et al 1981, not actually recent. However, we have proceeded to reformulate the paragraph.

Before, these lines read:

However, more recent studies using moorings and ship data have large ranges at lower estimates: 20-56 Sv in Fahrbach et al. (1991) and about 30 Sv in Yaremchuk et al. (1998).

Now, these lines read:

However, subsequent studies using moorings and ship data have large ranges at lower estimates: 20-56 Sv in Fahrbach et al. (1991) and about 30 Sv in Yaremchuk et al. (1998).

**Line 297: which ones? The ones above?**

Not only the above studies but also those studies listed in Table 4 of the submitted manuscript. We have proceeded to reformulate the paragraph.

Before, these lines read:

A handful of papers have addressed the Weddell Gyre's transport using idealized models, reanalysis products and observations.

Now, these lines read:

A handful of papers have addressed the Weddell Gyre's transport using idealized models, reanalysis products and observations, as listed in Table 4 (see section 3.3).

**Line 298: you mention 3 . I am confused.**
**Line 298-301: This has been mentioned various times now.**

We agree, we have removed the indicated text for clarity.

About the three different products, just to make it clear, as we explained in response to line 245 we use three global ocean products in the submitted manuscript. Two of these products are two NEMO-based products with the same forcing, and differing in their horizontal and vertical resolutions. The third product is HYCOM-based (GLBv0.08), which has different forcing and horizontal and vertical resolutions as compared to the NEMO-based products.

**Line 303-304: It wouldn't be necessary to remind the reader if the simulations were explained in detail when you show the results. I will give more suggestions on this in the general comment.**

Agree, we have removed the sentence as the information is already provided in the data section.

**Line 303-306: This is all information that should have been provided in the data description.**

We agree and have moved theisdescription of the datasets to section 2.1.

**Line 308-311: Repetition. The title of the subsection already indicates what it is about.**

We agree and have removed the repetitive paragraph.

**Line 313: In the next paragraph you write Figure 3. Please check for consistency throughout.**

Thanks, we realized about the formatting mistake, and have proceeded to check the entire manuscript for consistency.

**Line 313: you have numbered your panels. can you make use of it here? If you were to number the subplots vertically you could refer to Figure 3 a-d.**

Agree, we have proceeded to number the subplots and refer to them in the text.

**Line 315: again the last simulation does not match with the others with respect to the timeline. I wonder how these data are comparable.**

We appreciate your concern about the comparability but, while it is true that the two models have different lengths of data records, our comparison is based on climatological averages rather than year-by-year data. Climatological comparisons are standard practice in oceanographic research to identify long-term patterns and trends, and they are robust as the three reanalysis products cover more than 20 years of data. Therefore, we believe that the data are indeed comparable for the purposes of our study. However, we recognize that the inclusion of GLBv0.08 might have broadened the scope of analysis beyond the intended focus of our study, potentially exceeding a suitable number of pages.

For this reason, we have decided to omit results from the HYCOM-based product in the revised version of the manuscript to ensure clarity.

**Line 316-320: very long sentence. Split in two.**

Agree, we have splitted the sentence.

Before, these lines read:

The most prominent feature derived from altimetry data (geostrophic velocity field), and in all reanalysis products (total velocity) is the multi-jet structure of the WBCS, although less prominent in GLORYS2V4, and consisting of a series of parallel-aligned jets (CC, ASF, WF) running nearly perpendicular to the transect E-ADELIE, departing from the Antarctic Peninsula's tip and oceanward (Figure 3).

Now, these lines read:

The most prominent feature derived from altimetry data (geostrophic velocity field) and all reanalysis products (total velocity) is the multi-jet structure of the WBCS, although it is less prominent in GLORYS2V4. This structure consists of a series of parallel-aligned jets (CC, ASF, WF) running nearly perpendicular to the transect E-ADELIE, departing from the Antarctic Peninsula's tip and extending oceanward (Fig. 3).

**Line 322: and do they all agree? are there differences between what you found and what other studies have found?**

We have rephrased this sentence, in order to make it clear.

Before, these lines read:

This multi-jet structure has been previously reported from both observational works based on in situ observations (Muench & Gordon, 1995; Thompson & Heywood, 2008) and modelling studies (Stewart & Thompson, 2016; Matano et al., 2002).

Now, these lines read:

This multi-jet structure is in good agreement with both observational works based on in situ observations (Muench & Gordon, 1995; Thompson & Heywood, 2008) and modelling studies (Stewart & Thompson, 2016; Matano et al., 2002).

**Line 324-325: this is the 3rd time this has been stated.**

We agree and have removed the sentence.

**Line 326: do you think this could also be bathymetry related?**

We have been working with these reanalysis products in the Bransfield Strait (BS) for some months now. The tests we have been performing reveal that the most likely cause of the CC not entering into the strait is the atmospheric forcing since the entire surface circulation exits the BS following the westerlies.

Thus, we hypothesize that this issue is primarily due to the wind forcing in the model. However, elucidating the cause of this mismatch with the real ocean is out of the scope of the present work and we prefer not to make any statement about it beyond reporting the fact that the CC does not enter into the BS. We agree the bathymetry might play a role as well, but again, confirming this requires further investigation, which falls out of the present study.

**Line 335-336: Can you elaborate more on why that would be?**

Between lines 393 and 402, when we discuss the results, we explain that altimetry data have been reported to underestimate direct velocity measurements and why, providing scientific references.

Lines 393-402 of the submitted manuscript:

The weaker strength of these currents and jets in the modelled data as compared to observations might be attributed to two factors. On the one hand, the different time-scales involved in the comparison: the SADCP data are synoptic measurements and the modelled data are climatological values obtained after a time-averaging process (always expected to be smoother values). On the other hand, the three products assimilate remotely-sensed observations, including scatterometer and altimetry data, which help the products to adjust numerical solutions to measured surface ocean currents. If we account that remotelysensed derived surface currents (geostrophic plus Ekman currents) have been reported to underestimate direct velocity measurements, for instance by 27% on average in the Agulhas Current System but ranging by 4-64% (Hart-Davis et al., 2018), one can reasonably understand the magnitude offset in the reanalysis product data. Furthermore, regions rich in high mesoscale variability such as eddies, current meandering and instability waves are expected to result in simulation errors, especially when major currents are not represented by strong flows (Hewitt et al., 2020).

**Line 344-346: and why do you hypothesize that? I am missing a reasoning here. From Fig 3 I would be hesitant to assume that the currents are not topographically bound.**

We hypothesize this because there is neither a frontal structure accompanying the IWC area nor apparent topographical conditioning, i.e. the IWC spans over the (flat) deep sea ocean.
To support the hypothesis of the IWC being mainly driven by the wind stress forcing, we present Fig. R6 showing the mean vertical sections for the E-ADELIE transect, displaying conservative temperature, absolute salinity, and velocity, averaged from 1993 to 2020, for the GLORYS12V1 product. In this figure, the different branches are marked with black dashed lines. It can be observed that the IWC is a deep ocean interior branch with bathymetry around 4000m depth. Additionally, this figure highlights the frontal

structures in temperature and salinity at the positions of the CC, ASF, and WF (as defined in Heywood et al., 2004: https://doi.org/10.1029/2003JC002053); no such a structure is present in the IWC area.

Furthermore, we performed correlations between the seasonal cycles of the volume transport of each WBCS jet and the wind stress curl (Curlz) averaged for the area indicated in Fig. R4. Importantly, the near-to-the-coast branches (CC, ASF, and WF) show a lower influence by the Curlz seasonality ($R^2 = 0.3$, $R^2 = 0.3$, $R^2 = 0.5$), than the IWC does, showing a high correlation ($R^2 = 0.8$; see Table R4). This increasing correlation towards the ocean interior suggests that the jets closer to the continental shelf are less influenced by the wind stress forcing than the IWC is, and that the stronger influence of sea-ice formation/melting might be more relevant for jets closer to the continental shelf (CC, ASF and WF) as compared to the more oceanward IWC.

Before, these lines read:

We hypothesize this branch of the WBCS of the Weddell Gyre is mainly subject to the basin-scale wind-driven interior recirculation of the gyre and, subsequently, less influenced by topographic steering and thermohaline effects of the dense water formation along the continental slope.

Now, these lines read:

After performing correlations between the seasonal cycles of the volume transport driven by each current and the wind stress curl of the Weddell area (see section 2.2.3), we find an $R^2$ of 0.8 for the IWC while lower $R^2$ values are found for the CC ($R^2 = 0.3$), ASF ($R^2 = 0.3$) and the WF ($R^2 = 0.5$). Following this, we hypothesize that the wind forcing is the main driver behind the IWC, which is subject to the basin-scale interior recirculation of the gyre and is less influenced by topographic steering (the IWC flows away of the continental slope) and thermohaline effects from dense water formation along the continental slope (there is no evidence of any thermohaline front around the IWC; see vertical sections in Fig. A1).

**Line 356-357: speculation**

We have removed the results and discussion related to the HYCOM-based product so these lines are not present in the revised version.

**Line 361-364: I see that this is important information, but this clearly belongs to your description of data sets rather than your results.**

We think this information is important to be here, as the reader should take into account the origin and characteristics of the data to properly address the discussion section.

**Line 411-414 (about fig. 4): can you remind the reader of what the abbreviations stand for in your figure caption. The figure caption should be self-explanatory. also how are these current defined? Simply by velocities? I don't think so. what about water mass charcteristics.**
**can you merge subplots a and c and increase the map size? I can barely read the numbers on the given transects. subplots b and d can also be merged. also Fig 4. does not have a numbering of the subplots which are however referred to in the figure caption.**

We agree and have proceeded to rewrite the caption and modify the figure as suggested (see Fig. R7 at the end of the document).

Regarding the water mass characteristics of each branch, we have added a detailed table in the appendix (Table R1 and R2) in response to comments on lines 68-69. This table includes the main water masses encountered along the E-ADELIE transect, providing a comprehensive overview of their properties. Also, we would like to remind that the primary focus of this article is on the dynamics of the Western Boundary Current System (WBCS), rather than the hydrography of the area. While velocities are a key factor in defining and driving the currents, we hope that the table in the appendix ensures now that water mass characteristics are also sufficiently addressed for a more comprehensive understanding.

Before, these lines read:

Figure 4. Panels a and c show the bathymetry of the north-western sector of the Weddell Gyre depicting the area studied during two oceanographic campaigns in different seasons and years, corresponding to: (a) LG0003a of R/V Laurence M. Gould in April 2000, and (c) NBP0106 cruise of R/V Nathaniel B. Palmer in November 2001. The transects are shown in gross grey lines, and the area studied on this work is highlighted in black. Red dots indicate the distance along the transects. Panels b and d picture the corresponding velocity field (cm/s) of the cruises on the left, coming from SADCP measurements in the domain of E-ADELIE. Depth-averaged SADCP measurements (25-55 m) are presented as original data in grey, smoothed data in black. We do not indicate the ASF in panel (d) as a closer inspection to the current direction indicates a reversed flow over this location.

Now, these lines read:

Figure 4. (a) Bathymetry of the north-western sector of the Weddell Gyre, showing the area studied during two oceanographic campaigns in different seasons and years. The campaigns correspond to LG0003a of R/V Laurence M. Gould in April 2000 (black) and NBP0106 of R/V Nathaniel B. Palmer in November 2001 (blue). Transects are shown as fine lines, and the studied area is highlighted with thick lines. Black dots indicate the distance along the transects. (b) Corresponding velocity field (cm/s) for the tracks of the cruises shown in panel (a), derived from Shipboard Acoustic Doppler Current Profiler (SADCP) measurements in the E-ADELIE domain. Depth-averaged SADCP measurements (25-55 m) are presented as fine lines (original data) and thick lines (smoothed data). The four branches of the Western Boundary Current System are indicated: Coastal Current (CC), Antarctic Slope Front (ASF), Weddell Front (WF), and Inner Weddell Current (IWC). Note that the ASF appears to be absent in the NBP0106 cruise, and a closer inspection of the current direction indicates a reversed flow at this location.

**Line 419: I dont understand**

We have reformulated the caption for clarity (see the response above, Line 411-414).

**Line 420: how and why were the data smoothed. why have the SADCP data been averaged from 25-55m what is the reasoning behind this?**

The data were smoothed to highlight mesoscale and large-scale structures by reducing the influence of submesoscale processes, which are often present as high-frequency oscillations. We achieved this by smoothing the data every 10 km.

Regarding the averaging range of 25-55 m, the rationale is to enable a meaningful comparison between Fig. 4 and Fig. 3 of the submitted manuscript, which includes altimetry data representing surface measurements. By averaging the SADCP data over this depth range, we aim to capture near-surface currents while enhancing data reliability by including several depth bins, thereby increasing robustness in the measurements.

**Line 425: This is not relevant here. It has already been explained that the data were rotated.**

Agree, we have removed this sentence.

**Line 428: I dont understand.**

When we say 'as anticipated in Fig.3, we meant that the horizontal maps of the velocity field shown in Fig.3 already suggested that the multijet structure of the WBCS is absent in GLORYS2V4.

Before, these lines read:

This multi-jet structure is absent in GLORYS2V4 and mostly visible in GLORYS12V1 and GLBv0.08 (Fig. 5a, c, e), as anticipated in Fig. 3.

Now, these lines read:

This multi-jet structure is absent in GLORYS2V4 and more apparent in GLORYS12V1 (Fig. 5a, c), as one could reasonably expect from the horizontal distribution of the velocity field shown in Fig. 3, where GLORYS2V4 shows the ASF and WF as an unique major jet, while GLORYS12V1 clearly defines the ASF and WF as two distinct jets.

**Line 431-432: you refer to what it shows first and then the figure.**

Agree, we have reformulated the sentence as follows:

Before, these lines read:

… as previously noticed from the surface view in Fig. 3 (altimetry and reanalysis data) and Fig. 4 (SADCP measurements).

Now, these lines read:

… as previously noticed from the surface view of the altimetry and reanalysis data (Fig. 3) and the SADCP measurements (Fig. 4).

**Line 441: can you ad wich figure and which subplots you are referring to?**

These lines refer to a piece of text which has been removed from the revised version since it compared GLORYS and HYCOM-based products.

**Line 445: replace ',' to 'and', and 'and' to 'but'**

We have replaced these words in the revised version.

**Line 448-449: why showing it then?**

We have removed GLBv0.08 from the revised version.

**Line 455: different slopes of what? what is the first one? the slope? Are you describing a Figure here? please put reference.**

This text belongs to the discussion of results related to the HYCOM-based product and is not present in the revised version.

**Line 456-459: I do not understand what you are trying to say. Is that a comparison with Thompson and Heywood (2008). Or did you simply use a 'similar' approach but no comparison?**
**you did not include bottom triangles, in what sense is this method similar to Thompson and Heywood (2008).**
**for the different type of analysis 'similar' to Thompson and Heywood? I dont see what the difference between these approaches is. can you please elaborate?**

We think that here our writing could lead to a misunderstanding of the analyses performed, so we have reformulated these sentences and you can read them in the next response, which is linked to this comment.

Regarding the bottom-triangle calculations, we have updated all our calculations applying extrapolation towards the bottom, and added the results and associated explanations in the response to the question of line 258.

**Line 460: you have aleardy said that in the sentence before?**
**Line 461: define similar**
**Line 462: Replace ':' by 'of'**

When using the term 'similar', we are talking about our E-ADELIE transect but shorter (400 km offshore). The idea here is just to make a fair comparison between the increment of cumulative volume transport per km between our reanalysis data, and the data coming from the paper of Thompson and Heywood (2008). In the aforesaid paper the length of the ADELIE transect is almost 400 km, so that we take the same distance for our calculations. It is a comparison we perform based on results from Thompson and Heywood (2008) in their Fig. 11.

We have reformulated sentences for clarity taking into account the comments of the previous lines (from line 453 to 463).

Before, these lines (451-463) read:

Finally, the oceanward distribution of the cumulative transport, V', within E-ADELIE displays a common pattern where transport values increase markedly between a distance of 150 km and 470 km offshore in GLORYS2V4 and GLORYS12V1, and between 150 km and 600 km offshore in GLBv0.08. This increase is nearly linear in GLORYS2V4 and GLORYS12V1 and about 0.075 Sv per km and 0.076 Sv per km, respectively. Differently, in GLBv0.08 we distinguish two segments of different slopes for this transport increase. The first one is found to be between 150 km and 350 km and about 0.064 Sv per km; the second one is found to be between 350 km and 600 km and about 0.156 Sv per km. If we perform a similar analysis to find the rate of volume transport increase by distance but now based on the direct velocity measurements reported in Thompson and Heywood (2008), our calculations yield a rate of 0.11 Sv per km (derived from transport estimates in their Fig. 11). This rate of volume transport increase approaches our estimates based on reanalysis products. Following the cumulative transport shown in Thompson and Heywood (2008) along the ADELIE transect, their Fig. 11, the authors report approximately 46 Sv. Our estimates for a similar offshore transect length about 400 km lead to relatively lower transport estimates: 24.41 Sv for GLORYS2V4, 23.45 Sv for GLORYS12V1, and 19.38 Sv for GLBv0.08.

Now, these lines read:

Finally, the oceanward distribution of the cumulative transport per km, V', within E-ADELIE displays a common pattern where transport values increase markedly between a distance of 150 km and 470 km offshore in GLORYS2V4 and GLORYS12V1. This increase is nearly linear and about 0.075 and 0.076 Sv per km, respectively. If we perform the same analysis to find the rate of cumulative volume transport increase by distance but now based on the direct velocity measurements reported in Thompson and Heywood (2008), our calculations yield a higher rate of 0.11 Sv per km (derived from transport estimates in their Fig. 11). This rate of volume transport increase approaches our estimates based on reanalysis products. Moving to the total cumulative volume transport along the ADELIE transect, Thompson and Heywood (2008) report approximately 46 Sv, while our estimates for the E-ADELIE transect but shortened to 400 km length (same as ADELIE), lead to lower volume transport estimates of 27.8 Sv for GLORYS2V4 and 25.6 Sv for GLORYS12V1.

**Line 463: you are saying that the difference in these results occurs because you time averaged the reanalysis data yes?**
**What about other factors you did not consider, such as bottom triangles, different bathymetries and grid size (resolution) for the models or the potential of stronger interannual variability rather than seasonal variability?**
**I also cannot follow with respect to what exactly has been time-averaged here. Thompson and Heywood (2008) used observational data so they only had a limited time frame to consider. You on the other hand have years and years of data. did you use the same day and year to compare these transports or fully averaged our entire time series? Please elaborate.**

We appreciate your insights and concerns regarding the comparison between our reanalysis data and the observational data from Thompson and Heywood (2008). We have addressed the issue of bottom gaps to

ensure a more accurate comparison. However, as explained in our response to major comments, our primary aim is to highlight the structural similarities rather than directly compare the exact values between the products and observations. We acknowledge that a direct comparison between observations and reanalysis products is inherently challenging due to various factors such as different vertical and horizontal resolutions, grid sizes, or potential interannual variability. These elements can all contribute to discrepancies in the results. To clarify this, we have used climatological averages over the entire time series for our reanalysis data, rather than comparing specific days or years to the observational data from Thompson and Heywood (2008). We would like to note that this approach smooths the results and provides a more generalized view, which is a standard practice in oceanographic research as we mentioned in major comments. We will ensure that the discussion highlights these limitations and makes the readers aware of the inherent challenges in such comparisons. Taking these considerations into account, we have reformulated the text.

Before, these lines read:

We attribute these lower values in the reanalysis products, among other factors previously discussed, to their climatological nature compared to the synoptic LADCP measurements in Thompson and Heywood (2008).

Now, these lines read:

We attribute these lower values in the reanalysis products to several factors, including different vertical and horizontal resolutions (as discussed previously), their climatological nature compared to the synoptic LADCP measurements in Thompson and Heywood (2008), and the inherent challenges in directly comparing observations with reanalysis products. Nevertheless, the main aim is to highlight that both the reanalysis products and observational data fall within similar ranges of transport and exhibit similar jet structures.

**Line 480 (Fig. 5): have you considered doing an EOF on this ? It would be really interesting to see where the variability is strongest and which variability dominates with respect to cross-transect transports (interannual or seasonal variability)**
**the axis labels are very small, can you increase the fontsize?**

We appreciate your suggestion regarding the EOF analysis. While our current focus is on the seasonal cycle, and we plan to address interannual variability in future research, we recognize the value of your suggestion. Therefore, we conducted an EOF analysis to explore this aspect further with GLORYS12V1 data. We present the results in Fig. R8.

The results indicate that the WF and ASF play a major role in the variability of the WBCS compared to other currents. However, although these findings are insightful, we prefer to keep them to be included in an ongoing manuscript about the interannual variability shown by these reanalysis products. We think that for the present study those results are beyond the scope of the present work.

Regarding the font size, we show a new figure in response to comment of line 258 (Fig. R1).

**Line 480: Replace 'as modelled in' to 'as provided in'**

Agree, these words have been replaced.

**Line 485: the E-ADELIE transect**

Agree, this word has been corrected.

**Line 485-486: whereas bottom reaching means different depths for the reanalysis with different resolutions.**

We agree with your observation. As we mentioned earlier, we applied a bottom-reaching method to standardize calculations across all products, ensuring consistency.

**Line 489: why?**

We extended DL farther offshore to 600 km to capture the cumulative transport area of the GLBv0.08 product, as shown in Fig. 5f. In this figure, the cumulative V' extends farther offshore compared to the NEMO-based products. This extension ensures that we account for the full V' of the IWC branch in the GLBv0.08 product, providing a more accurate representation of the dynamics in this model.

However, we recall that, for the sake of clarity, we have decided to exclude GLBv0.08 from our revised analysis, so we will no longer discuss this product.

**Line 498-500: and why would you expect that?**

We appreciate the reviewer's question regarding our expectations about the rate of cumulative volume transport. In the original manuscript, we observed that the cumulative volume transport rate increased significantly with distance offshore in the GLBv0.08 product, indicating different dynamics in the second area (350-600 km offshore) compared to the first area (150-350 km offshore).

However, as we have decided to exclude GLBv0.08 from our revised analysis, we will no longer discuss this product. Instead, we will focus on a detailed comparison and discussion of the GLORYS products to ensure clarity and consistency in our study.

**Line 502: of cross-transect transport yes?**
**what are the values? Can you give a range?**

We have proceeded to add the notion 'cross-transect' to provide context to the transport estimates, and also to provide actual numbers to provide values for the different seasons.

Before, these lines (502-503) read:

In Fig. 6, the three panels display a clear seasonal cycle with maximum values through autumn-to-winter and minimum values through summer-to-spring, …

Now, these lines read:

In Fig. 6, the three panels display a clear seasonal cycle with maximum values of cross-transect transport through autumn-to-winter (~40 Sv) and minimum values through summer-to-spring (~35 Sv), …

**Line 503: Fig. 6a same for the other figure references**

We agree and have proceeded to check the entire manuscript for consistency.

**Line 508: replace 'follow each other closely' to 'mostly align'**

Agree, we have replaced these words.

**Line 508-509: compared to what?**

As we said in the main text, we are comparing GLORYS2V4 against GLORYS12V1. However, we have proceeded to reformulate this for clarity in response to the next comment.

**Line 515: well you talk about the seasons (the range you defined before), but then you only refer to specific months. what is the seasonal average?**

In the main text, we mention the months with minimum and maximum transport, which correspond to summer and winter, respectively. We are not only identifying seasons but also specifying the particular months. However, to address your comment, we provide the seasonal averages for clarity. The seasonal average values are presented in response to lines 54-55.

We also proceed to reformulate the text to ensure this distinction is clear in the revised version, taking into account the decision of excluding GLBv0.08 product.

Before, these lines (508-520) read:

In Fig. 6a, the seasonal cycles from GLORYS2V4 and GLORYS12V1 follow each other closely, being always about 2 Sv higher in GLORYS12V1 through the months transitioning to/from maxima values. Both products agree on peak volume transports reaching 32-33 Sv in autumn and winter months. The minima appear in January at 24 Sv and 26 Sv, respectively, for GLORYS2V4 and GLORYS12V1. This leads to a seasonal amplitude of about 7-8 Sv. In GLBv0.08, volume transport estimates computed for the same transect length (DL between 150 km and 470 km offshore), lead to a more conspicuous seasonal cycle with a maximum mean transport of 37 Sv in July (winter) and minimum mean transport of 21.5 Sv in December (summer), leading to a larger seasonal amplitude about 15.5 Sv, as compared to GLORYS products. This larger seasonal amplitude occurs because GLBv0.08 seems to overestimate (underestimates) the seasons 515 with highest (lowest) transports. When computing the volume transport in GLBv0.08 for the longer section (DL between 150 km and 600 km and accounting for the ASF, WF and broader IWC), we refer to this as GLBv0.08*. Then, the maximum and minimum transports occur through the same months as in GLBv0.08 but now reach 56 Sv and 48 Sv, respectively (decreasing its seasonal amplitude of variation down to 8 Sv). The latter suggests that most of the WBCS seasonal variability in GLBv0.08 is embedded between 150 km and 470 km distances offshore the Antarctic Peninsula.

Now, these lines read:

In Fig. 6a, the seasonal cycles from GLORYS2V4 and GLORYS12V1 follow each other closely, being about 2 Sv higher in GLORYS2V4 through the months transitioning to/from maxima values. Both products agree on peak volume transports reaching 42.2 ± 1.3 and 39.5 ± 0.8 Sv in winter months, and 42.6 ± 1.6 and 40 ± 0.6 Sv in autumn, respectively, for GLORYS2V4 and GLORYS12V1. The minima appear in summer showing 33.5 ± 3.1 Sv and 33.8 ± 3.0 Sv, followed by spring values (37.1 ± 4.0 and 36.1± 2.8 Sv), respectively. This leads to a seasonal amplitude of about 7-10 Sv, depending on the product.

**Line 525: you are plainly describing the figures here. where is your discussion? is this comparable to previous studies, what are the caveats of the differnet reanalysis products. Why are they showing different values for the cross-transect transport. There is so much more to discuss here. Simply saying a specific model underestimates something because of resolution is simply not enough. Seasonal cycles are clearly different with the different considered depth ranges. What does this tell you.**

We agree R#1 and have expanded the discussion to provide a deeper analysis. We address these points with the following additional paragraph after Line 526 in the submitted manuscript.

New additional paragraph (after Line 526):

We hypothesize that the discrepancies between both products are primarily due to differences in grid resolution. Both products utilize the same forcing; however, the lower resolution in GLORYS2V4 results in the absence of the multi-jet structure of the WBCS, showing the ASF and WF as a broad single branch. This is evident in Fig. 3d and f, where GLORYS2V4 displays higher values in the domain between 100-300 km offshore E-ADELIE compared to GLORYS12V1. This accounts for the higher transport values in the seasonal cycle integration for GLORYS2V4. Regarding the vertical resolution, previous studies (Neme et al, 2021; Renner et al, 2009; Styles et al, 2023) have shown that model resolution plays a significant role in accurately representing oceanic structures and currents in the Weddell Gyre, especially for bottom-intensified jets. The differences in seasonal cycles between the considered depth ranges suggest that the transport dynamics are sensitive to vertical stratification and jet structure, which are better resolved in

higher resolution models like GLORYS12V1. This highlights the importance of using high-resolution models for detailed oceanographic studies and suggests that the GLORYS2V4's coarser resolution may smooth out critical features, leading to underestimation or misrepresentation of transport values.

**Line 525: they dont follow each other. They align or they are similar...**

Agree, we have changed the words in the main text following the reviewer's suggestion.

**Line 530: how? NCEP CFS shows a much larger wind stress than compared with ERA5/ ERAInterim**

In the main text of the manuscript, we state that "it becomes evident that the three panels predominantly portray the same pattern." Here, we are referring to the overall pattern and seasonal cycle, rather than specific values. All panels depict a similar pattern with maximum and minimum values occurring in the same months, highlighting the consistency in seasonal variation across the datasets, despite differences in the magnitude of wind stress.

However, we will not further discuss any issue related with GLBv0.08, following the decision to exclude it from the revised version.

**Line 540: you haven't caluclated the wind stress curl?**

We agree with your suggestion to calculate the wind stress curl. To ensure clarity, we also take into account the presence of Sea Ice when calculating wind stress and wind stress curl, as suggested by R#2. We have added the results of these calculations in Table R3, which will be also a part of the revised version.

The explanations about the wind stress curl computation are detailed in response to the comment for Line 286; see further above in this same document of response).

**Line 560: longer E-ADELIE transect**

It is not a longer E-ADELIE transect, it is a longer section within the E-ADELIE transect (extending 150-600 km offshore, and presented in Table 2 of the submitted manuscript). However, we will not further discuss this since results from the HYCOM-based product (GLBv0.08) have been removed from the revised version.

**Line 560-561: prevents further discussion of the results.**

This line refers to results based on the HYCOM-based product; accordingly, these lines have been removed from the revised version.

**Line 595-596: WBCS**
**Line 602-603: you have already defined the abbreviation for this. You should summarize your results and not spend space on redefining those things.**
**Line 604-605: have you considered looking at the impacts in hydrographic characteristics downstream?**
**Line 616: yes bottom-intensified jets at depth and then you did not consider to include bottom triangles to add to you transport calculations? Also bathymetry is significantly differnet for both these products and you looked at completely different time averages.**
**Line 617: what are the characteristics of these fronts. how did you define them?**
**Line 619-620: you really haven't talked about it much. Is it seasonal? its is continuously there? is it always in the same location?**
**Line 639-641: to what extent ? What is needed to improve the results.**
**Line 641-644: In what sense? Which reanalysis provides the best results and why?**

Regarding the comments above highlighted in yellow font, we must note they all belong to the conclusions sections, and so they do not exist anymore as they were in the submitted version. We will take all these minor suggestions into account when drafting the final conclusions and will be provided once the Discussion Forum is closed and we are allowed to upload the revised version of the manuscript.

Comments in Line 616 and Line 617 are already addressed among previous responses on this document (see responses to Lines 68-69, 411-414, and Tables R1 and R2).

**Below you can find the new references of the revised version:**

Chad A. Greene, Kaustubh Thirumalai, Kelly A. Kearney, José Miguel Delgado, Wolfgang Schwanghart, Natalie S. Wolfenbarger, Kristen M. Thyng, David E. Gwyther, Alex S. Gardner, and Donald D. Blankenship. The Climate Data Toolbox for MATLAB. Geochemistry, Geophysics, Geosystems 2019. doi:10.1029/2019GC008392                    https://doi.org/10.1029/2019GC008392.

Heywood, K. J., A. C. Naveira Garabato, D. P. Stevens, and R. D. Muench, On the fate of the Antarctic Slope Front and the origin of the Weddell Front, J. Geophys. Res., 109, C06021, doi:10.1029/2003JC002053, 2004.

Kwok, R., & Cunningham, G. F. (2015). Variability of Arctic sea ice thickness and volume from CryoSat-2. Philosophical Transactions of the Royal Society A: Mathematical, Physical and Engineering Sciences, 373(2045), 20140157. https://doi.org/10.1098/rsta.2014.0157.

John D'Errico (2024). inpaint_nans (https://www.mathworks.com/matlabcentral/fileexchange/4551-inpaint_nans), MATLAB Central File Exchange. Retrieved June 17, 2024.

Lüpkes, Christof, and Gerit Birnbaum. "Surface drag in the Arctic marginal sea-ice zone: a comparison of different parameterisation concepts." Boundary-layer meteorology 117.2 (2005): 179-211.

Smith, W. H., & Sandwell, D. T. (1997). Global sea floor topography from satellite altimetry and ship depth soundings. *Science*, *277*(5334), 1956-1962.

**Below you can find the new figures/ tables of the revised version:**

[Figure]

Figure R1. (a, c) Time-averaged velocity field of v' (cm/s) for E-ADELIE as provided in GLORYS2V4 and GLORYS12V1, respectively. (b, d) Cumulative transport, V' (Sv) of the time-averaged velocity field of v' for E-ADELIE as provided in GLORYS2V4 and GLORYS12V1, respectively. The time-average corresponds to the period between 1993-2020. Dashed black lines are used as reference for discussion of the results in the text. Black dots at 650 km offshore represent the vertical resolution of each product.

| Water mass | Conservative temperature (°C) | | Salinity | | Depth (m) | | Density (Kg/m³) | |
|---|---|---|---|---|---|---|---|---|
| Data | Bibliography | GLORYS12V1 | Bibliography (PSU) | GLORYS12V1 (g/kg) | Bibliography | GLORYS12V1 | Bibliography Neutral density | GLORYS12V1 Potential density |
| AASW | -1.9~0 | -2~1.5 | 33.8~34.50 | 32.50~34.50 | 0~250 | 0~200 | ≤ 28.00 | ≤ 27.60 |
| HSSW | - | -1~-1.9 | - | 34.40~34.70 | - | 150~500 | - | 27.60~27.80 |
| WDW | 0~0.6 | 0.2~0.8 | 34.65~34.69 | 34.60~34.70 | 200~2000 | 200~2000 | 28.10~28.27 | 27.80~27.85 |
| WSDW | -0.7~0 | -0.2~0.2 | ~34.65 | 34.70~35.10 | 1500~4000 | 500~3000 | 28.27~28.40 | 27.85~27.86 |
| WSBW | -1.4~0 | -0.2~-1 | 34.60~34.65 | ~34.70 | ≥ 1000 | ≥ 1000 | ≥ 28.40 | ≥ 27.86 |

Table R1. Characteristics of the main water masses encountered along the E-ADELIE transect as seen in bibliography (Thompson and Heywood, 2008) and in GLORYS12V1 reanalysis product (averaged from 1993 to 2020). The table includes the key water masses for this region: Antarctic Surface Water (AASW), High Salinity Shelf Water (HSSW), Warm Deep Water (WDW), Weddell Sea Deep Water (WSDW) and Weddell Sea Bottom Water (WSBW).

| Current | Conservative temperature (°C) | | Salinity | | Potential density (Kg/m³) | |
|---------|-------------------------------|------------|----------------------|---------------------|----------------|------------|
| Data | *Bibliography* | *GLORYS12V1* | *Bibliography (PSU)* | *GLORYS12V1 (g/kg)* | *Bibliography* | *GLORYS12V1* |
| *CC* | -2~0 | -1.6~-0.6 | 34.35~34.61 | 33.63~34.83 | - | 26.92~27.83 |
| *ASF* | -2~ 1 | -1.6~0.21 | 34.35~34.65 | 33.93~34.84 | 27.80~28.40 | 27.17~27.85 |
| *WF* | 0~0.6 | -1.6~0.4 | 34.65~34.70 | 34.02~34.86 | ≥ 28.00 | 27.25~27.86 |
| *IWC* | - | -1.7~0.4 | - | 34.08~34.86 | - | 27.30~27.87 |

Table R2. Characteristics of the main water masses encountered along each jet of the WBCS as seen in bibliography (Heywood et al, 2004) and in GLORYS12V1 reanalysis product (averaged from 1993 to 2020). The table includes currents described on the E-ADELIE transect: CC (Coastal Current), ASF (Antarctic Slope Front), WF (Weddell Front) and IWC (Inner Weddell Current).

[Figure]

Figure R2. Bathymetric map of the north-western sector of the Weddell Gyre depicting the study area with an indication to the approximate location of major oceanographic features and currents with a surface signal. The E-ADELIE transect localization is indicated in blue (see section 2.2.1). Acronyms stand for: Southern Boundary of Antarctic Circumpolar Current (ACC), Western Boundary Current System (WBCS), Antarctic Coastal Current (CC), Antarctic Slope Front (ASF), Weddell Front (WF) and Inner Weddell Current (IWC).

[Figure]

Figure R3. Bathymetric map of the north-western sector of the Weddell Gyre depicting the study area with indication of the ADELIE transect (Cruise 158 of RRS James Clark Ross (JCR) in February 2007) studied in Thompson & Heywood (2008), and the E-ADELIE transect we study. The ADELIE stations are indicated with black dots; the station numbers are also shown. Distance of E-ADELIE is indicated along the transect. The new frame of reference for the velocity field is also indicated (the Cartesian coordinates were rotated 15.79 degrees clockwise (α) from the true east following Eq. 1). Fine white lines trace bathymetric contours at 200- and 400-meters depth, while grey lines correspond to depths of 1000, 2000, 3000, 4000, and 4500 meters.

[Figure]

Figure R4. Horizontal map of the wind stress (unitary vectors) and wind stress curl (N/m3) for the Weddell Sea area calculated between 1993 and 2020. The area of integration for wind stress and wind stress curl is highlighted in red-dashed lines. The E-ADELIE transect is marked in black solid line.

[Figure]

Figure R5. (a) Monthly climatology of the volume transport, V', across E-ADELIE with their corresponding standard deviation (shades around the climatological values). The coloured thick lines (see legend) refer to each product as follows: grey solid line (GLORYS2V4), black solid line (GLORYS12V). Observational data for analogous transects to E-ADELIE reported in the literature, and listed in Table 4, are plotted in coloured dots when corresponding to measurements taken at a synoptic scale, and in horizontal dashed lines when corresponding to a time-averaged estimate. (b) Same as panel above, but computed for the surface layer (from 0 to 100 m depth). (c) Basin-scale seasonal wind stress (black) and wind stress curl (grey) seasonal cycles in Pa and N/m3, respectively. Wind stress curl is multiplied by 10^6 to match the range of wind stress for plotting purposes. The area of integration is specified in the appendix (Fig. R4)., computed over the study region.

| Reanalysis product | R$^2$ between surface V' and wind-stress | R$^2$ between surface V' and wind-stress curl | R$^2$ between total V' and wind-stress | R$^2$ between total V' and wind-stress curl |
|---|---|---|---|---|
| GLORYS2V4 | 0.848 | 0.852 | 0.804 | 0.733 |
| GLORYS12V1 | 0.833 | 0.879 | 0.762 | 0.719 |
| GLBv0.08 | 0.836 | 0.831 | 0.700 | 0.595 |

Table R3. Results derived from linear model regression, where y represents the total and surface cross-transect volume transports in each scenario, and x denotes the wind stress and wind stress curl. All reported values exhibit p-values approaching 0 and are presented with 95% coefficient bounds.

| Jets | Product | GLORYS2V4 | GLORYS12V1 |
|---|---|---|---|
| CC | | 0.32 | 0.29 |
| ASF | | 0.39 | 0.54 |
| WF | | 0.51 | 0.42 |
| IWC | | 0.83 | 0.81 |

Table R4. R$^2$ after correlating seasonal curlz and seasonal volume transport of each jet. Wind stress curl calculation takes into account the presence of Sea Ice as suggested R#2. Note that GLORYS2V4 and GLORYS12V1 are averaged from 1993 to 2020.

[Figure]

Figure R6. Vertical sections showing Conservative Temperature, Absolute Saltinity and v' (cm/s) mean conditions (1993-2021) of each branch as depicted in GLORYS12V1 product. The position of the fronts is defined following the black dashed lines and their names, corresponding to: Coastal Current (CC), Antarctic Slope Front (ASF), Weddell Front (WF), Inner Weddell Current (IWC).

[Figure]

Figure R7. (a) Bathymetry of the north-western sector of the Weddell Gyre, showing the area studied during two oceanographic campaigns in different seasons and years. The campaigns correspond to LG0003a of R/V Laurence M. Gould in April 2000 (black) and NBP0106 of R/V Nathaniel B. Palmer in November 2001 (blue). Transects are shown as fine lines, and the studied area is highlighted with thick lines. Black dots indicate the distance along the transects. (b) Corresponding velocity field (cm/s) for the tracks of the cruises shown in panel (a), derived from Shipboard Acoustic Doppler Current Profiler (SADCP) measurements in the E-ADELIE domain. Depth-averaged SADCP measurements (25-55 m) are presented as fine lines (original data) and thick lines (smoothed data). The four branches of the Western Boundary Current System are indicated: Coastal Current (CC), Antarctic Slope Front (ASF), Weddell Front (WF), and Inner Weddell Current (IWC). Note that the ASF appears to be absent in the NBP0106 cruise, and a closer inspection of the current direction indicates a reversed flow at this location.

[Figure]

Figure R8. EOF analysis of the velocity field displayed in Fig. R1 as depicted for GLORYS12V1 product.

---

## Community Comment (CC4)

Dear Reviewer #2,

We appreciate the time and efforts you have dedicated in providing feedback about our manuscript and are grateful for your insightful comments. These comments have contributed to improve our research work. We have now finalized the revision of our manuscript entitled "On the Seasonal Western Boundary Current System of the Weddell Gyre".

In the following, a detailed point-by-point response to comments by reviewer #2 is presented. To make a clear distinction, comments from the reviewer #2 are marked in bold font while our response is in regular font. To ease their identification through this document, the new text for the revised version of the manuscript is highlighted in blue font.

Please note that the lines indicated as LXXX in our response refer in all cases to the submitted manuscript. Please, note that new figures and tables mentioned in our response are presented at the end of this document. These new figures and tables will be indicated as Figure RY or Table RY, where R stands for Revision and Y stands for an ascending number. If we refer to Figures and Tables of the submitted manuscript, we will use the same format as in the submitted manuscript.

Before proceeding with the detailed point-by-point response, we provide below an overview of the main changes applied in the revised version to ease its assessment:

1.- After performing the analyses suggested by the reviewers, and carefully addressing their concerns jointly, we find their suggestions add so many important aspects to the story that the space left for discussion of differences among NEMO and HYCOM led us to decide to drop the latter from the revised version. We are happy the first submitted version shows the major differences among products, but we agree that maybe we were trying to cover too much at once and by doing so the main message may be unclear. The revised paper focuses on NEMO-based products and observations (direct velocity measurements and altimetry data) solely. We use this variety of data to characterize the seasonal variations of the Western Boundary Current System of the Weddell Gyre assessing an open-access product against existing observations and former modeling studies in order to enable its future use in studies about the interannual variability of this current system. Also, we describe the ocean dynamics governing the interior branch of the ocean gyre, demonstrating this is highly controlled by the basin-wide wind stress forcing as opposed to the most relevant role that thermohaline forcing plays as one approaches the coastal zone (sea-ice formation/melting dominates here).

*Following the suggestion by #Reviewer1: to drop the HYCOM-based results from the manuscript.*

2.- We have recomputed the wind stress-forcing in all cases accounting for the presence of sea-ice making use of the algorithms develop by Greene et al., 2019:

Chad A. Greene, Kaustubh Thirumalai, Kelly A. Kearney, José Miguel Delgado, Wolfgang Schwanghart, Natalie S. Wolfenbarger, Kristen M. Thyng, David E. Gwyther, Alex S. Gardner, and Donald D. Blankenship. The Climate Data Toolbox for MATLAB. Geochemistry, Geophysics, Geosystems 2019. doi:10.1029/2019GC008392 https://doi.org/10.1029/2019GC008392.

*Following the suggestion by #Reviewer2: to account for the sea-ice presence when computing wind stress.*

Also, we have included in our analysis correlations of the volume transport and the wind stress curl.

*Following the suggestion by #Reviewer1: to account for the basin-wide wind forcing acting over the ocean gyre.*

3.- We have filled in the gaps near the bottom-sea accounting for the bottom-intensified jets. To do so we have used an algorithm available in MATLAB code that we have previously used successfully in Veny et al. (2022) to perform a similar exercise where data near an island slope needed to be filled prior to volume transport calculations. This algorithm* accounts for the surrounding gradients to perform a realistic extrapolation. We will discuss this method in the revised version as well as include a mention of the bottom-triangle approach described by Thompson and Heywood (2008).

John D'Errico (2024). inpaint_nans (https://www.mathworks.com/matlabcentral/fileexchange/4551-inpaint_nans), MATLAB Central File Exchange.

*Following the suggestion by #Reviewer1: to account for the gaps near the sea bottom.*

We look forward to hearing back from you.

Sincerely,

Tania Pereira-Vázquez
* * *
POINT-BY-POINT RESPONSE

**REVIEWER #2**

**This presented manuscript investigates the Wendell Gyre's western boundary current system structure using three different reanalysis products with addition of in-situ hydrographic data and altimetry data. The main motivation is to assess the capability of open-access reanalysis products in producing relevant dynamics and variability of the system. The authors, highlight the existence of a bottom reaching, broad current extending up to 600km offshore eastward of the Weddell Front, which they name Inner Weddell Current.**

**The southern hemisphere subpolar gyres constitute an important role in our climate systems by modulating the poleward heat transport and ultimate supply of heat that drives basal melting of the Antarctic ice shelves. On the other hand, the western boundary current system transports transformed water masses northward where they can participate in the global thermohaline circulation. Given the challenges in observing, in particular remote and ice-coverage regions, gridded reanalysis datasets are a popular and tempting choice to do data analysis. However, a careful validation is needed, thus this study has potential to be a valuable contribution to the community's understanding of the regional dynamics and variability as well as providing insights about the performance of existing broadly used reanalysis products.**

**Overall, the manuscript is written clearly and presented figures support the main conclusions of the work. However, I have several comments and concerns about this work as presented in more detail below. I recommend rejection with encouragement of resubmission.**

We thank reviewer #2 for the careful reading of our work and for the time and efforts dedicated to providing feedback about our research work. Now, we proceed to the point-by-point review.

**Major comments**

**It would be better to not refer to the section as ADELIE section but actually as SR04 WOCE. To my understanding ADELIE refers to the specific project/cruise in 2007, which steamed along the WOCE SR04 transect to release drifters. If the authors want to keep their current terminology this should at least be mentioned, as actually also done in Thompson & Heywood (2008).**

We appreciate the comment and will clarify this information regarding the transect in the revised version of the manuscript.

We have decided to maintain our current terminology, referring to it as the E-ADELIE transect. However, we agree with R#2 that it is important to mention the similarity with the SR04 WOCE section. Therefore, we will rephrase the submitted manuscript to specify that the E-ADELIE transect is a portion of the existing SR04 WOCE section. This will ensure clarity and proper context for our readers while retaining the terminology we have used throughout the manuscript.

Before, these lines (237-246) read:

2.2.1 The E-ADELIE transect
To characterize the variability of the WBCS of the Weddell Gyre we focus on a key location, the historical ADELIE transect (Fig. 2), where the current system is well defined and has been previously described as a multi-jet structure current system (Thompson & Heywood, 2008). The ADELIE transect is located northeast of the AP, where the Weddell waters may either turn around the Antarctic Peninsula towards the Bransfield Strait, leave the gyre circulation towards the Scotia Sea and the South Atlantic Ocean, or recirculate within the Weddell Gyre. Different from the traditional ADELIE transect, we analyse a version of that one which extends farther oceanward into the gyre interior and which we name E-ADELIE transect. This transect aims to account for the dynamics of the western branch of the Weddell Gyre to its full extent. We address the comparison of the horizontal and vertical structure of the WBCS jets as depicted by the two reanalysis products, SADCP measurements and altimetry data.

Now, these lines read:

**2.2 Methods**

To characterize the variability of the WBCS of the Weddell Gyre, we focus on a key location, the western part of the historical SR04 WOCE section. The western portion of this section was previously studied by Thompson & Heywood (2008), who refer to this section as the ADELIE transect (Fig. 2), where the current system is well defined and described as a multi-jet structure current system. Different from the ADELIE transect, we analyze a version that extends farther oceanward into the gyre interior, which we name the E-ADELIE transect. Here, E-ADELIE refers to the western portion of the SR04 WOCE section. This transect aims to capture the dynamics of the western branch of the Weddell Gyre in its entirety, including the IWC jet. We address the comparison of the horizontal and vertical structure of the WBCS jets as depicted by the two reanalysis products (GLORYS2V4 and GLORYS12V1), SADCP measurements, and altimetry data.

**Wind stress calculation: wind stress in ice covered regions is altered by presence of sea ice, so in order to get the actual surface stress for the ocean, sea ice should be considered. Overall, the discussion of sea ice and how it mediates momentum transfer at the ocean surface is missing. It is concerning to me if authors are not aware of this process.**

We agree with R#2 and have revised our wind stress and wind stress curl calculations to account for Sea Ice Coverage, utilizing the toolbox developed by Greene et al. (2019):

Chad A. Greene, Kaustubh Thirumalai, Kelly A. Kearney, José Miguel Delgado, Wolfgang Schwanghart, Natalie S. Wolfenbarger, Kristen M. Thyng, David E. Gwyther, Alex S. Gardner, and Donald D. Blankenship. The Climate Data Toolbox for MATLAB. Geochemistry, Geophysics, Geosystems 2019. doi:10.1029/2019GC008392                                          https://doi.org/10.1029/2019GC008392.

By integrating these adjustments into our analysis, we ensure that the impact of sea ice on wind stress and its implications for momentum transfer at the ocean surface are properly considered. We have also updated the methodology section to include these considerations.

Before, these lines (264-291) read:

We calculate wind stress as follows, using the formula proposed by Kara et al. (2013):
$$\tau = \rho \cdot U_{10}^2 \cdot C_D \text{ (3)}$$
where $\rho$ represents the air density (1.2 kg m-3); $U10 = \sqrt{u10'2 + v10'2}$ is the wind speed at 10 m above the surface (with v10' and v10' denoting rotated eastward and northward velocity components, respectively); and, CD is the drag coefficient, which is a function of wind speed, U10.
To ensure alignment with volume transport calculations, we compute cross- and along-ADELIE wind velocities. v10' and u10' are utilized to calculate U10, maintaining the same rotation angle ($\alpha$) through the expressions:
$$v_{10}' = v_{10} * cos(\alpha) - u10 * sin(\alpha) \text{ (4)}$$
$$u_{10}' = v_{10} * sin(\alpha) + u10 * cos(\alpha) \text{ (5)}$$
We conduct basin-scale calculations to assess the average wind patterns across our study area. Subsequently, we analyse the relationship between the seasonal cycles of wind and volume transport,

considering wind as a potential influencing factor. To do this, we delineate the boundaries of the study are as follows: the northern boundary corresponds to a distinct change in wind direction at 64ºS; moving southward, the limit is established by identifying the point where the signal of the WBCS becomes evident, which occurs at 74ºS; the western boundary is demarcated by the Antarctic Peninsula (AP) at 62ºW, while the eastern boundary is determined by an observable shift in both wind and current directions at 38ºW. The products employed for wind stress computation derived from the two reanalysis products used in this study. GLORYS2V4 and GLORYS12V1 use ERA-interim and ERA5 datasets, respectively. Since ERA-interim was discontinued in 2019, ERA5 forcing fields have been applied starting from January 2019, accessible at https://www.ecmwf.int/en/forecasts/dataset/ecmwf-reanalysis-interim and https://cds.climate.copernicus.eu/cdsapp#!/dataset/reanalysis-era5-single-levels?tab=overview. GLBv0.08 uses NCEP-CFS and NCEP-CFSV2 products. NCEP-CFS was discontinued in 2010. The corresponding data sets are accessible at https://www.hycom.org/dataserver/ncep-cfsr and https://www.hycom.org/dataserver/ncep-cfsv2.

Now, these lines read:

We calculate wind stress as follows, using the formula proposed by Lüpkes et al. (2005):

$$\tau = \rho \cdot U_{10}^2 \cdot C_i \qquad (2)$$

where $\rho$ represents the air density (1.2 kg m$^{-3}$); $U_{10} = \sqrt{(u'^2_{10} + v'^2_{10})}$ is the wind speed at 10 m above the surface (with $u'_{10}$ and $v'_{10}$ denoting rotated eastward and northward velocity components, respectively); and, $C_i$ is the drag coefficient calculated following Lüpkes et al. (2005), after specifying the sea ice concentration.

Wind stress curl is calculated after obtaining the $\tau_x$ and $\tau_y$ components of the wind stress, accounting for sea ice over the Weddell Basin. Basin-scale calculations are performed to assess the average wind patterns across our study area (see Fig. YY in the appendix). We then analyze the relationship between the seasonal cycles of wind and volume transport, considering wind as a potential influencing factor.

**The authors should give some clarification about the satellite altimetry product used. This seems to be the standard sea level product used in the global open ocean. My understanding is that the traditional gridded altimeter products cannot be readily used in ice covered regions. It might be useful to at least also use the Dynamic topography and sea level anomaly product by Armitage et al., 2018. The authors mention as well in L123-126 that remote sensing products have caveats during the ice-covered season, however they do not further discuss this in relation to the data they use for the analysis or the discussion. This is another concern, leaving me wondering how careful the authors were in their analysis and choice of data.**

The Weddell Sea is characterized by harsh winter conditions, with extensive sea-ice coverage for most of the year except summer. This presents a significant challenge for the use of traditional altimetry products, particularly during the winter season. The altimetry product we utilized in this study is detailed in the 'Quality Information Document' (Ref: CMEMS-SL-QUID-008-032-068, https://doi.org/10.48670/moi-00148). This document stipulates a baseline of 15% sea ice concentration to flag data, following Lavergne et al. (2019). In addition, for our analysis, we only used altimetry data when at least 50% of the dataset along the E-ADELIE transect contained valid values, resulting in the absence of winter season data. Note also that the altimetry data shown in Fig. 3 represents a seasonal composite of thirty years of data. Moreover, our choice and careful handling of data are consistent with previous studies in the Weddell Sea area and Bransfield Strait, such as:

*Oelerich, R., Heywood, K. J., Damerell, G. M., du Plessis, M., Biddle, L. C., & Swart, S. (2023). Stirring across the Antarctic Circumpolar Current's southern boundary at the prime meridian, Weddell Sea. Ocean Sci., 19, 1465–1482. https://doi.org/10.5194/os-19-1465-2023.*

*Frey D, Krechik V, Gordey A, Gladyshev S, Churin D, Drozd I, Osadchiev A, Kashin S, Morozov E and Smirnova D (2023). Austral summer circulation in the Bransfield Strait based on SADCP measurements and satellite altimetry. Front. Mar. Sci. 10:1111541. https://doi.org/10.3389/fmars.2023.1111541.*

Before, lines (238-246) read:

The surface geostrophic circulation of the study area is derived from SEALEVEL_GLO_PHY_L4_MY_008_047 (Global Ocean Gridded L4 Sea Surface Heights and Derived Variables Reprocessed 1993 Ongoing), hereafter ALT. This product is derived from various altimeter missions and encompasses data from GEOSAT to Jason-3. The altimeter data is processed using the DUACS multimission altimeter data processing system. The spatial resolution is 0.25° × 0.25°, with a temporal resolution of daily covering the period from 1993 to 2020.

Now, these lines read:

The surface geostrophic circulation of the study area is derived from SEALEVEL_GLO_PHY_L4_MY_008_047 (Global Ocean Gridded L4 Sea Surface Heights and Derived Variables Reprocessed 1993 Ongoing), hereafter ALT. This product is derived from various altimeter missions and encompasses data from GEOSAT to Jason-3. The altimeter data is processed using the DUACS multimission altimeter data processing system. The spatial resolution is 0.25° × 0.25°, with a temporal resolution of daily covering the period from 1993 to 2020.

Since the Weddell Sea is characterized by harsh winter conditions, with extensive sea-ice coverage for most of the year except summer, the use of traditional altimetry products becomes a significant challenge. The altimetry product we employ accounts for the presence of sea-ice coverage, as indicated in the quality information document (CMEMS-SL-QUID-008-032-068). This document indicates that a baseline of 15% sea ice concentration is used to flag data, following Lavergne et al. (2019). Furthermore, we established a threshold criterion requiring a minimum of 50% spatial coverage for the altimetry data along the E-ADELIE transect to be included in our analyses. Consequently, this results in the exclusion of winter season climatology due to insufficient data coverage.

**The authors base their description of the new current (Inner Weddell Current) on two SADCP sections. I believe there are more SADCP sections that could be used, e.g. https://doi.pangaea.de/10.1594/PANGAEA.735277 Furthermore, the altimetry product has to be treated with great caution as this region is partially ice covered all year round, even in summer.**

We appreciate the comment about the SADCP data. We have analyzed more than ten SADCP sections in this region. The decision to include the two transects presented in the manuscript was based on their comprehensive coverage of the study transect across different years and seasons (the campaigns correspond to LG0003a of R/V Laurence M. Gould in April 2000 and NBP0106 of R/V Nathaniel B. Palmer in November 2001). We think this approach helps to support the presence of the Inner Weddell Current (IWC) across different temporal contexts.

We consider the objective of the study is not to provide a rigorous analysis of all the existing SADCP data around the study transect but to compare the main features of the Western Boundary Current System against some observational data. If the reviewers consider necessary to show additional SADCP transects, we can do so as a part of the Appendix.

Regarding the altimetry product, we hope the explanation provided in the previous point raised by R#2 was clear henough.

**I am not 100% convinced that it is necessary to name what the authors refer to as Inner Weddell Current if it basically is the western part of the gyre circulation. Maybe there is another clear distinction, which I am not picking up. If that is the case, it would be great to specifically highlight that distinction. The authors state themselves in L619-620 "… seems to drive largely the recirculation of interior water within the gyre", which, again, to me is just the western branch of the gyre circulation and does not qualify as a 'newly discovered' current.**

We think it is important to keep the definition of the inner circulation as IWC since this is not simply the western part of the gyre circulation. The WBCS of the Weddell gyre is composed of a multi-jet structure rather than by one unique branch. Each current has been shown to display its own spatial pattern and downstream recirculation (as discussed in the submitted manuscript); not every jet follows the same fate. Downstream of the ADELIE transect, the CC recirculates around the Antarctic Peninsula, and the ASF and

WF split into different branches (some leaving the basin and some recirculating within the gyre) while the IWC belongs to the inner recirculation of the gyre where flows are shown not to leave the basin (Fig.3). We think it is important to account for all these branches separately, especially when the IWC is shown to carry significantly different amounts of volume transport as compared to the other jets (Lines 466-477 of the submitted manuscript).

**Minore comments**

**A map with the full Weddell Gyre would be helpful in introduction. Maybe the one already shown can be then used as an inset**

We agree with R#2 and, in line with comments from R #1 (Line 41, Line 77), we have modified Fig. 1 to ensure a better representation of the study area (see Fig. R1 at the end of this document). However, we prefer to maintain the study area as the main map instead of an inset. This approach ensures that the dynamics of the area remain clear. The revised Fig. 1 will now include a map showing the full Weddell Gyre, providing a comprehensive overview while keeping the detailed study area prominent.

**Careful in comparing snap-shots with time mean averages of products of different length**

We appreciate this important point raised by Reviewer #2 and also by Reviewer #1 (Major comments). Direct comparisons between observations and reanalysis products are inherently challenging due to the limitations in both datasets. Observational data often do not reach the bottom in all stations, and reanalysis products vary in their temporal coverage and resolution. To address these concerns, we will highlight these limitations in our discussion and the comparison of the estimates derived from reanalysis products and those reported in the literature. While a straightforward comparison may not be possible, we think the results remain valuable and fall within the expected range of volume transport values. Furthermore, this is a traditional approach followed by former studies (Matano et al., 2002; Neme et al., 2021, their Table 1) where estimates from modelling or reanalysis products are compared with existing observations, with a clear acknowledgment of the limitations involved. In the revised version we make the readers aware of these limitations and provide a careful discussion of our findings.

**Figures:**

**Add isobaths for orientation (e.g. in Figure 3)**
**Figure 3: It would be helpful to add some markers for distance or mark the individual fronts, described in the right side panels, in the maps. To me the jets are not readily visible on the surface fields**

We agree with your suggestion and have added isobaths and distance markers along the E-ADELIE section for better orientation (see Fig R2).

**Line-based comments:**

**L33: Grammar: '…approaches the real ocean … the closest when compared…' does not seem correct**

We agree. In the revised version we have replaced these expressions.

Before, these lines read:

Results from this study suggest that the high-resolution version of NEMO (GLORYS12V1) approaches the real ocean in the western Weddell Sea the closest when compared to observations and literature.

Now, these lines read:

Results from this study suggest that the high-resolution version of NEMO (GLORYS12V1) better captures the major features of the western Weddell Sea when compared to observations and literature, namely the

location and spatial distribution of the multi-jet structure and the volume transport driven by each current of the WBCS.

**L61: Not just a leakage of subsurface but also surface waters? It could be relevant to refer to Morrison et al., 2023 here.**

We agree and have rephrased this line: The CC drives the exit of Weddell waters towards Bransfield Strait, allowing the leakage of near-freezing surface subsurface waters (Morrison et al., 2023).

**L70: known as WDW within the Weddell Sea. Mention that CDW enters the gyre at its eastern boundary (e.g. Schroeder et al., 1999 and Ryan et al., 2016)**
**L69/70: ' Down the continental slope…' this statement is not correct in my opinion. Down the slope AABW is found, or potentially refer to Weddell Sea Deep and Bottom Water. High salinity shelf water is formed due to Brine rejection on the Filchner Ronne shelf and the Larsen shelf, which is then exported locally and flows down the continental slope.**

We agree with this point, previously raised by R#1, and we proceed to better explain the process occurring in the area.

Before, these lines (65-70) read:

Across this front there is a rapid change in temperature and salinity due to the interaction between the colder and fresher waters of the Antarctic continental shelf, formed as a result of sea ice formation and melting processes, and the relatively warm and saline waters sourced by the Antarctic Circumpolar Current (ACC) (Thompson & Heywood, 2008; Vernet et al., 2019). The former water mass is known as Antarctic Surface Water (AASW), while the latter water mass is a modified Circumpolar Deep Water known as Warm Deep Water (WDW). Down the continental slope, dense Antarctic Bottom Water (AABW) is formed following brine rejection in the upper ocean surface.

Now, these lines read:

Across this front there is a rapid change in temperature and salinity due to the interaction between the colder and fresher waters of the Antarctic continental shelf, formed as a result of sea ice formation and melting processes, and the relatively warm and saline waters sourced by the Antarctic Circumpolar Current (ACC) entering the gyre from the eastern part (Thompson & Heywood, 2008; Vernet et al., 2019). The former water mass is known as Antarctic Surface Water (AASW), while the latter water mass is a modified Circumpolar Deep Water known as Warm Deep Water (WDW) within the Weddell Sea (Schröder & Fahrbach, 1999). Over the continental shelf, dense shelf water produced during sea ice formation cascades down the continental slope and mixes with WDW. This process allows the dense water to reach the ocean bottom and continue its pathway through the global ocean. This dense bottom water is known as Antarctic Bottom Water (AABW) (Muench & Gordon, 1995; Stewart & Thompson, 2012).

**L70-73: this is a very busy sentence which would profit from some citations, e.g. how does the ASF impact sea-ice dynamics and are there examples of impacting marine ecosystems?**

We agree with this point, previously raised by R#1, so we have proceeded to reformulate the sentences.

Before, these lines read:

Thus, the ASF plays a crucial role in the exchange of heat, salt, and nutrients between the deep ocean and the Antarctic continental shelf waters with important implications for the distribution of marine ecosystems and sea-ice dynamics, where the WDW also conditions the melting of ice shelves.

Now, these lines read:

Thus, the ASF plays a crucial role in the exchange of heat, salt, and nutrients between the deep ocean and the Antarctic continental shelf waters, with important implications for the distribution of marine ecosystems

and sea-ice dynamics (Vernet et al, 2018). In these regions, the WDW also influences the melting of ice shelves. However, the ASF is not entirely circumpolar; it is interrupted by the Antarctic Peninsula, separating the Pacific and Atlantic sectors of the Southern Ocean (Thompson et al., 2018). Specifically, it is absent along the West Antarctic Peninsula and large parts of the Bellingshausen Sea.

**L73: how is the Weddell Front defined? Could be helpful to mark the fronts in the hydographic sections in the appendix**

Following the suggestion of R#1, we have added a table in the appendix (see Table R1 and R2), clarifying the hydrographic characteristics of each jet:

Lines 75-76: An overview of the water masses which compose the multi-jet structure of the WBCS in the reanalysis products as compared to observations is provided in the Appendix, including their main characteristics (Fig. A1 and Table R1 and R2).

Appendix:

For clarity, we provide an overview of the characteristics of the main water masses encountered along the E-ADELIE transect in Table R1 and R2. This summary is based on the literature and data from the GLORYS12V1 product. These ocean property ranges are in agreement with those found for GLORYS2V4.

To address the reviewer's concern about the front's position, we have added a vertical dashed line marking the positions of the fronts (Fig. R3).

**L172/173: DOI should be moved to acknowledgements or Data Availability section, same for url of ERA-Interim**

We agree and have moved this information to the *Data Availability Section,* as required by the Journal.

All data we used in this investigation are publicly accessible. Global NEMO-based reanalysis products GLORYS2V4 and GLORYS12V1 are available at https://doi.org/10.48670/moi-00024 (last access 29/05/2024) and https://doi.org/10.48670/moi-00021 (last access 08/02/2024), respectively. Data of Sea Ice Coverage from GLORYS products are also accessible at the same link as the products (last access 13/06/2024). Regarding wind products, those forcing GLORYS products are ERA-interim and ERA5 datasets, which are available in open access at https://www.ecmwf.int/en/forecasts/dataset/ecmwf-reanalysis-interim (last access 29/05/2024) and https://cds.climate.copernicus.eu/cdsapp#!/dataset/reanalysis-era5-single-levels?tab=overview (last access 29/05/2024).
Regarding altimetry data, they are available at https://doi.org/10.48670/moi-00148 (last access 19/12/2022).
Both SADCP data are free accessible, LG0003a of R/V Laurence M. Gould cruise data are available at http://adcp.ucsd.edu/lmgould/lmgadcp/year2000.html#lg0003a (last access 14/02/2024), and NBP0106 cruise of R/V Nathaniel B. Palmer cruise data are available at https://currents.soest.hawaii.edu/nbpalmer/nbp0106/nb150/webpy/ (last access 29/02/2024).
Finally, in situ data from Cruise SOS-Climate II of RV Ary Rongel (Fig. A1 from appendix), are accessible at https://doi.pangaea.de/10.1594/PANGAEA.864578 (last access 26/05/2021).

**L529: so are ERA5 and ERA-interim the same line?**

Yes, we use the same forcing as specified in the technical documents of each product. We use ERA-Interim data until 2018 and ERA5 data from 2019 onwards. This is explained in the main text in section 2.2.3, Wind Stress.

**L530: What do you mean by the same pattern? That larger windstress in NCEP CFS would driver larger transport in GLBv0.08? This should be clarified in the text.**

In the revised version we have removed the HYCOM-based product following the suggestions by reviewer R1 and our own argumentation provided at the beginning of this document of response.

**L540: citation should be Le Paih 2020**

We thank the reviewer for pointing out this. We have corrected the citation to reference Le Paih (2020).

**L628: Try to avoid subjective verbiage**

Thank you for your suggestion. We have revised the entire document accordingly. However, if when providing the revised document, you still note specific instances where you feel the language could be improved, we will be happy to address the requested changes.

**References**

**Armitage, Thomas W. K., Ron Kwok, Andrew F. Thompson, and Glenn Cunningham. 2018. "Dynamic Topography and Sea Level Anomalies of the Southern Ocean: Variability and Teleconnections." Journal of Geophysical Research, C: Oceans 123 (1): 613–30.**

**Morrison, Adele K., Matthew H. England, Andrew Mcc Hogg, and Andrew E. Kiss. 2023. "Weddell Sea Control of Ocean Temperature Variability on the Western Antarctic Peninsula." Geophysical Research Letters 50 (15): e2023GL103018.**

**Below you can find the new references of the revised version:**

Chad A. Greene, Kaustubh Thirumalai, Kelly A. Kearney, José Miguel Delgado, Wolfgang Schwanghart, Natalie S. Wolfenbarger, Kristen M. Thyng, David E. Gwyther, Alex S. Gardner, and Donald D. Blankenship. The Climate Data Toolbox for MATLAB. Geochemistry, Geophysics, Geosystems. doi:10.1029/2019GC008392, 2019.

Heywood, K. J., A. C. Naveira Garabato, D. P. Stevens, and R. D. Muench, On the fate of the Antarctic Slope Front and the origin of the Weddell Front, J. Geophys. Res., 109, C06021, doi:10.1029/2003JC002053, 2004.

John D'Errico (2024). inpaint_nans (https://www.mathworks.com/matlabcentral/fileexchange/4551-inpaint_nans), MATLAB Central File Exchange. Retrieved June 17, 2024.

Kwok, R., & Cunningham, G. F. Variability of Arctic sea ice thickness and volume from CryoSat-2. Philosophical Transactions of the Royal Society A: Mathematical, Physical and Engineering Sciences, 373(2045), 20140157. https://doi.org/10.1098/rsta.2014.0157, 2015.

Lüpkes, Christof, and Gerit Birnbaum. "Surface drag in the Arctic marginal sea-ice zone: a comparison of different parameterisation concepts." Boundary-layer meteorology 117.2: 179-211, 2005.

Morrison, Adele K., Matthew H. England, Andrew Mcc Hogg, and Andrew E. Kiss. "Weddell Sea Control of Ocean Temperature Variability on the Western Antarctic Peninsula." Geophysical Research Letters 50 (15): e2023GL103018, 2023.

Smith, W. H., & Sandwell, D. T. Global sea floor topography from satellite altimetry and ship depth soundings. *Science*, *277*(5334), 1956-1962, 1997.

**We add here new figures/ tables of the revised version:**

[Figure]

Figure R1. Bathymetric map of the north-western sector of the Weddell Gyre depicting the study area with an indication to the approximate location of major oceanographic features and currents with a surface signal. The E-ADELIE transect localization is indicated in blue (see section 2.2.1). Acronyms stand for: Southern Boundary of Antarctic Circumpolar Current (ACC), Western Boundary Current System (WBCS), Antarctic Coastal Current (CC), Antarctic Slope Front (ASF), Weddell Front (WF) and Inner Weddell Current (IWC).

[Figure]

Figure R2. Panels a, c, e, present the horizontal structure of the WBCS of the Weddell Gyre following the time-averaged velocity field (cm/s) at surface. The displayed arrows represent unitary vectors. E-ADELIE is indicated with a black line, and white dots over the line are presented every 100 km. Fine white lines trace bathymetric contours at 200- and 400-meters depth, while white gross lines correspond to depths of 1000, 2000, 3000, 4000, and 4500 meters. Panels b, d, f, depict the seasonal surface velocity field (cm/s) along E-ADELIE with its standard deviation. In panel (b), the winter season is absent, based on the requirement that at least more than 50% of the dataset along the E-ADELIE contains not empty values. Computations encompass the complete time series of each dataset for all panels.

| Water mass | Conservative temperature (°C) | | Salinity | | Depth (m) | | Density (Kg/m³) | |
|---|---|---|---|---|---|---|---|---|
| Data | Bibliography | GLORYS12V1 | Bibliography (PSU) | GLORYS12V1 (g/kg) | Bibliography | GLORYS12V1 | Bibliography Neutral density | GLORYS12V1 Potential density |
| AASW | -1.9~0 | -2~1.5 | 33.8~34.50 | 32.50~34.50 | 0~250 | 0~200 | ≤ 28.00 | ≤ 27.60 |
| HSSW | - | -1~-1.9 | - | 34.40~34.70 | - | 150~500 | - | 27.60~27.80 |
| WDW | 0~0.6 | 0.2~0.8 | 34.65~34.69 | 34.60~34.70 | 200~2000 | 200~2000 | 28.10~28.27 | 27.80~27.85 |
| WSDW | -0.7~0 | -0.2~0.2 | ~34.65 | 34.70~35.10 | 1500~4000 | 500~3000 | 28.27~28.40 | 27.85~27.86 |
| WSBW | -1.4~0 | -0.2~-1 | 34.60~34.65 | ~34.70 | ≥ 1000 | ≥ 1000 | ≥ 28.40 | ≥ 27.86 |

Table R1. Characteristics of the main water masses encountered along the E-ADELIE transect as seen in bibliography (Thompson and Heywood, 2008) and in GLORYS12V1 reanalysis product (averaged from 1993 to 2020). The table includes the key water masses for this region: Antarctic Surface Water (AASW), High Salinity Shelf Water (HSSW), Warm Deep Water (WDW), Weddell Sea Deep Water (WSDW) and Weddell Sea Bottom Water (WSBW).

| Current | Conservative temperature (°C) | | Salinity | | Potential density (Kg/m³) | |
|---------|---------|---------|---------|---------|---------|---------|
| Data | Bibliography | GLORYS12V1 | Bibliography (PSU) | GLORYS12V1 (g/kg) | Bibliography | GLORYS12V1 |
| CC | -2~0 | -1.6~-0.6 | 34.35~34.61 | 33.63~34.83 | - | 26.92~27.83 |
| ASF | -2~ 1 | -1.6~0.21 | 34.35~34.65 | 33.93~34.84 | 27.80~28.40 | 27.17~27.85 |
| WF | 0~0.6 | -1.6~0.4 | 34.65~34.70 | 34.02~34.86 | ≥ 28.00 | 27.25~27.86 |
| IWC | - | -1.7~0.4 | - | 34.08~34.86 | - | 27.30~27.87 |

Table R2. Characteristics of the main water masses encountered along each jet of the WBCS as seen in bibliography (Heywood et al, 2004) and in GLORYS12V1 reanalysis product (averaged from 1993 to 2020). The table includes currents described on the E-ADELIE transect: CC (Coastal Current), ASF (Antartic Slope Front), WF (Weddell Front) and IWC (Inner Weddell Current).

[Figure]

Figure R3. Vertical sections showing Conservative Temperature, Absolute Saltinity and v' (cm/s) mean conditions (1993-2021) of each branch as depicted in GLORYS12V1 product. The position of the fronts is defined following the black dashed lines and their names, corresponding to: Coastal Current (CC), Antarctic Slope Front (ASF), Weddell Front (WF), Inner Weddell Current (IWC).